

**Cr(VI) reduction, electricity production, and microbial resistance variation in**
**paddy soil under microbial fuel cell operation**
Huan Niu[a], Xia Luo[a], Peihan Li[a], Hang Qiu[a], Liyue Jiang[a], Subati Maimaitiaili[a],
Minghui Wu[a], Fei Xu[b], Heng Xu[b], Can Wang[a]*
a Sichuan Engineering Research Center for Biomimetic Synthesis of Natural Drugs,
School of Life Science and Engineering, Southwest Jiaotong University, Chengdu,
610031, Sichuan, P.R. China
b Key Laboratory of Bio-Resource and Eco-Environment of Ministry of Education,
College of Life Sciences, Sichuan University, Chengdu 610065, Sichuan, PR China
* Corresponding at No. 111 Second Ring Road, Chengdu, Sichuan, 610031, China. Tel:

11     + 86 18728156952

E-mail: wangcan@swjtu.edu.cn (Can Wang)



**Abstract figure:**

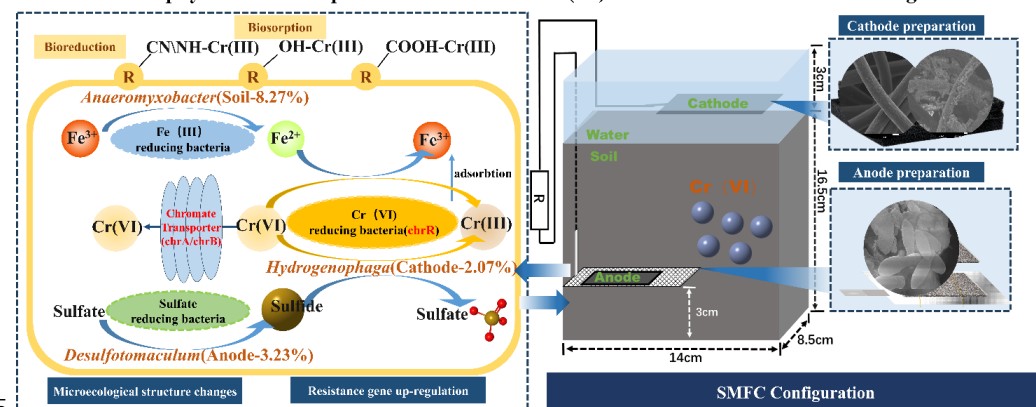

**Abstract:** Microbial fuel cell (MFC) is an efficient in-situ approach to combat pollutants and generate
electricity. This study constructed a soil MFC (SMFC) to reduce Cr(VI) in paddy soil and investigate its
influence on microbial community and microbial resistance characteristics. $Fe_3O_4$ nanoparticle as the
cathodic catalyst effectively boosted power generation (0.97 V, 102.0 mW/m$^2$), whose porous structure
and reducibility also contributed to Cr reduction and immobilization. After 30 days, 93.67% of Cr(VI)
was eliminated. The bioavailable Cr decreased by 97.44% while the residual form increased by 88.89%.
SMFC operation greatly changed soil enzymatic activity and microbial structure, with exoelectrogens
like *Desulfotomaculum* (3.32% in anode) and Cr(VI)-reducing bacteria like *Hydrogenophaga* (2.07% in
cathode) more than 1000 folds of soil. In particular, SMFC operation significantly enhanced the
abundance of heavy metal resistance genes (HRGs). Among them, *chrA, chrB, and chrR* increased by
99.54~3314.34% in SMFC anode than control, probably attributed to the enrichment of potential
tolerators like *Acinetobacter, Limnohabitans,* and *Desulfotomaculum.* These key taxa were positively
correlated with HRGs but negatively correlated with pH, EC, and Cr(VI), which could have driven Cr(VI)



reduction. This study provided novel evidence for bioelectrochemical system application in contaminated
paddy soil, which could be a potential approach for environmental remediation and detoxification.
**Keywords:** Chromium; Microbial fuel cell; Microbial response; Metal resistance



| Nomenclature | |
|---|---|
| SMFC | Soil microbial fuel cell |
| HRGs | Heavy metal resistance genes |
| HMs | Heavy metals |
| HGT | Horizontal gene transfer |
| EABs | Electrochemical active bacteria |
| GF | Graphite felt |
| ORR | Oxygen reduction reaction |
| WCV | Working circuit voltage |
| OCV | Open circuit voltage |
| ARGs | Antibiotic resistance genes |
| CMFC-A | Anode of the closed circuit group |
| OMFC-A | Anode of the open circuit group |
| CMFC-C | Cathode of the closed circuit group |
| OMFC-C | Cathode of the open circuit group |
| CMFC-S | Soil of the closed circuit group |
| OMFC-S | Soil of the open circuit group |
| NMFC-S | Soil of the non-electrode control group |

## 1. Introduction


Chromium (Cr) is one of the main toxic heavy metals (HMs), which enters the environment mainly
due to its wide use in electroplating, tanning, and other industries (Coetzee et al., 2020). Mineral-sourced
phosphate fertilizer also contains high-level Cr, further promoting its spreading and migration in soil and
underground water (Chen et al., 2022a). Even a sub-dose of Cr (especially the Cr(VI) state) can promote
plasmid-mediated horizontal gene transfer (HGT) (Zhang et al., 2018), causing enrichment of heavy
metal resistance genes (HRGs), threatening environmental safety (Guo et al., 2021; Wang et al., 2023a;
Wang et al., 2023b; Wang et al., 2020a). Due to the cross-resistant effect of HMs and antibiotics, the
enrichment of HRGs and tolerators under Cr exposure has become an emerging concern.
Common remediation methods for Cr-influenced soil include chemical reduction and leaching
(Cong et al., 2022), electrokinetic remediation (Morales-Benítez et al., 2023), and phytoaccumulation
(Yaashikaa et al., 2022), which convert Cr(VI) into insoluble and low toxic forms (e.g., Cr(III)) by
adsorption, ion exchange, and redox (Rani et al., 2022). Among them, the microbial approach using
functional microbes is commonly used for the continuous treatment of soil-groundwater, which has a



low cost without side effects (Fan et al., 2023). However, pollutants can be tightly adsorbed by soil
particles and persistently remain (Wang et al., 2023a). The complex soil constituents and competition of
indigenous microorganisms inhibit the colonization and development of functional microbes and limit
their effectiveness (Guo et al., 2021).

Microbial fuel cell (MFC) technology can transform or immobilize HMs and generate electricity

utilizing electrochemical active bacteria (EABs) (Chen et al., 2022b; Gupta et al., 2023), which have
been used in sediment or soil to treat HMs and organics and monitor environmental toxicity (Li et al.,
2023b). At present, soil MFC (SMFC) has been used for pollution control, focusing on pollutant content
and forms as well as the electrochemical properties (Hamdan and Salam, 2023; Liu et al., 2023a). There
is a lack of systematic research about the MFC effect on soil microbial community structure shifting and
resistance characteristics.

In this study, an SMFC was constructed to remediate Cr(VI) contaminated paddy soil. EABs were

pre-loaded on the SMFC anode to promote electricity production and Cr transformation. Ferroferric
oxide ($Fe_3O_4$) nanoparticles, which can reduce and fix  Cr(VI) directly, were used as a catalyst for
cathodic oxygen reduction reaction (ORR) (Liu et al., 2023b). During operation, Cr(VI) was reduced and
immobilized by bio-physical adsorption and electrochemical-microbial reduction, simultaneously. The
Cr(VI) reduction mechanism was comprehensively studied along with the analysis of microbial
community structure shaping and HRG variation. For the first time, SMFC-driven Cr(VI) reduction was
associated with microbial resistance, which evolved along with microbial adaption and development.
This study not only provides a reference for the microbial remediation of polluted soil but also improves
the practical field application of MFC.



## 2.    Materials and    Methods
**2.1. Chemicals**
All the chemicals and reagents were analytical grade or premium pure from Kelong Chemical
Reagent Factory, Chengdu, China.
**2.2 Construction of SMFC**
2.2.1 Soil
Paddy soil from Jintang County, Chengdu, China (30°74' N, 104°59' W) was collected and used to
construct SMFC. The soil has organic matter, organic carbon, and total nitrogen of 8.84±0.02%,
1.74±0.01%, and 321.67±1.25 mg/kg, respectively. Potassium dichromate was added to the soil to
achieve a final Cr(VI) concentration of 118.8 mg/kg before use.
2.2.2 Electrodes Preparation
Aluminum foam (66.0×54.0×5.0 mm) (SANZHENG Metal material, Chengdu, China) was used
as anode. The anode microflora was derived from municipal sludge (Chengdu Sixth Sewage Treatment
Plant, China) after acclimating with 100 mg/L Cr(VI). Before assembling, the aluminum foam was
cultivated in the anode microflora for 2 weeks. Then the anode was tied to titanium mesh tightly with
titanium wire. Graphite felt (GF) (100.0×50.0×3.0 mm) was used as the cathode (Table S1). Before
use, it was cleaned, dried, and loaded with $Fe_3O_4$ as the ORR catalyst, as detailed in section 1 of the
supplementary material. For characterization, we utilized a scanning electron microscope (SEM) to
examine the structure and morphology of the electrode surface. In addition, we performed X-ray
photoelectron spectroscopy (XPS) and energy dispersive spectroscopy (EDS) to analyze the valence state



and element composition. The phase composition was determined using an X-ray diffractometer (XRD).
2.2.3 SMFC construction
A plastic box (140.0×85.0×165.0 mm) was used as the SMFC reactor, with 1.50 kg soil and
overlying water of 3.0 cm to simulate the flooded state during rice planting. The cathode was floated on
the water surface while the anode was buried (about 3.0 cm from the bottom). The cathode and anode
were connected to a 2000 Ω resistor using titanium wire. The water level was kept constant by daily
replenishment.
**2.3 Design and Operation**
Three treatments were set up as shown in Fig. S1. Three parallel was set for each group.
1. NMFC: The control group with no electrode. It only contains an equal amount of overlying water
and paddy soil.
2. OMFC: The open circuit group with disconnected electrodes and equal amounts of overlying
water and paddy soil.
3. CMFC: The complete closed-circuit SMFC capable of producing electricity, with electrodes
connected by a 2000 Ω resistor, and equal amounts of overlying water and paddy soil.
The experiments were conducted at 25℃. A Raspberry Pi data acquisition system (ARMv7
architecture) was connected at both ends of the resistor of CMFC to monitor the voltage. A multimeter
was used for verification. The detailed code information can be found in the supplementary material
(Section 2). Soil and water samples were taken every 5 days until day 35, and the operation continued
for another 10 days until day 45. The electrochemical properties of the SMFC including the polarization
curve and power density curve were tested using an electrochemical workstation on days 15 and 30



(Ch660e, Shanghai Chenhua Instrument Co., Ltd., Shanghai, China) (Chen et al., 2022b).
**2.4 Cr Migration and Transformation**
To determine total Cr, 0.50 g soil was subjected to acid digestion (HCl-HNO$_3$-HClO$_4$) before
measurement using flame atomic absorption spectrometry (FAAS) (PinAAcle 900T AA Spectrometer,
PerkinElmer, America). To determine Cr speciation, BCR sequential extraction was applied to divide Cr
into HOAc extractable, reducible, oxidizable, and residual fractions with mobility and availability from
high to low (Wang et al., 2020a). Also, Cr(VI) concentration in overlying water was determined by a
spectrophotometer at 540 nm, while Cr(VI) in soil was determined using FAAS after alkaline digestion
(Fan et al., 2021). Duplicates, method blanks, and standard reference materials were used for quality
control. Cr recovery in standard reference materials was 92~108%.
**2.6 Microbial response during operation**
2.6.1 Soil biochemical response
Soil dehydrogenase (DHA) activity was measured using 2, 3, 5-triphenyl tetrazolium chloride (TTC)
method. Urease activity was determined by the phenol sodium hypochlorite colorimetric assay. Invertase
activity was determined by the 3, 5-dinitro salicylic acid colorimetric assay. The acid phosphatase (ACP)
activity was determined by the p-nitrophenyl disodium phosphate colorimetric assay (Wang et al., 2019;
Wang et al., 2017).
2.6.2 Microbial community structure
The microbial community structure of the electrodes and soil was determined by high-throughput
sequencing. Majorbio (Shanghai, China) performed 16S rRNA gene sequencing using the Illumina HiSeq



platform. 0.50 g of fresh homogenized samples were used to extract the total bacterial DNA with a
universal DNA Kit (Omega Biotek Inc., USA). After amplification and purification, the V3-V4
hypervariable regions of the bacterial 16S rRNA gene were amplified with primer pairs 338F and 806R.
After sequencing, the operational taxonomic units (OTUs) with a 97 % similarity cutoff were clustered
using UPARSE version 7.1, and chimeric sequences were identified and removed. The taxonomy of each
OTU representative sequence was analyzed by RDP Classifier version 2.2 against the 16S rRNA database
using a confidence threshold of 0.7. The alpha diversity, beta diversity, microbial community structure
change, and environmental factor correlation analysis were conducted. (Wang et al., 2023c)
2.6.3 HRG Fluctuation
The abundance of HRGs in the surface soil of SMFC and OMFC anode after operation was analyzed
using an SYBR Green real-time fluorescence quantitative PCR system (7500, Thermo Fisher, USA)
(Wang et al., 2023a). The soil of OMFC was used for comparison. The detected genes included HRGs
(*chrA, chrB, chrR, recG, nfsA, zupT, fpvA*) and MGEs (*intI, tnpA02, tnpA04, tnpA05*). The primer
sequences are provided in Table S2.
**2.7 Data analysis**
The experimental data were evaluated using one-way analysis of variance (ANOVA) based on three
tests. The mean values and standard deviations were calculated using SPSS 22.0 software (IBM, USA).
A significance level of $P<0.05$ was considered statistically significant, while $P<0.01$ was defined as
highly statistically significant. Graphs were plotted using Origin 2022 software.



## 3. Results

### 3.1 Electrodes characterization



As demonstrated in Fig. S2, Fig. 1, and Fig. 2, the raw GF had smooth surfaces with C, and O as
the main elements (Fig. S2). After $Fe_3O_4$ loading, black patches constituted with spherical particles
appeared, bringing Fe (13.17%) and O (19.97%) on the GF surface (Fig. 1a-c and Table S3). XPS found
that peaks of 710.8 and 724.4 eV were consistent with typical $Fe_3O_4$ peaks (Fig. 2a-b), indicating its
successful loading. The CV curves of the cathode (Fig. 2c) presented an obvious oxidation peak at 0.85
V, indicating its excellent electrochemical performance.
After operation, the typical peaks of Cr(III), Cr(VI), and element Cr (576.1 and 578.92) were
observed on both electrodes by XPS (Fig. 2d-e), indicating the reduction and immobilization of Cr(VI)
by the electrodes. GF was found loaded with many soil elements including Cr, Na, Mg, and Ca (Table
S3). SEM also observed many microorganism cells and extracellular organic-like substances, implying
the biofilm formation on the cathode (Fig. 1d-f).
As presented in Fig. S2C, the raw aluminum foam showed a rough porous structure with mainly Al
and O on the surface (Table S3). After loading EAB, many spherical and rod-shaped bacteria were
observed, indicating a good capacity to carry microorganisms (Fig. 1g-i). After the operation, many
millimeter-scale soil particles were embedded in the anode interspace, indicating the intense mass
transfer between the anode and soil (Fig. 1j-l).



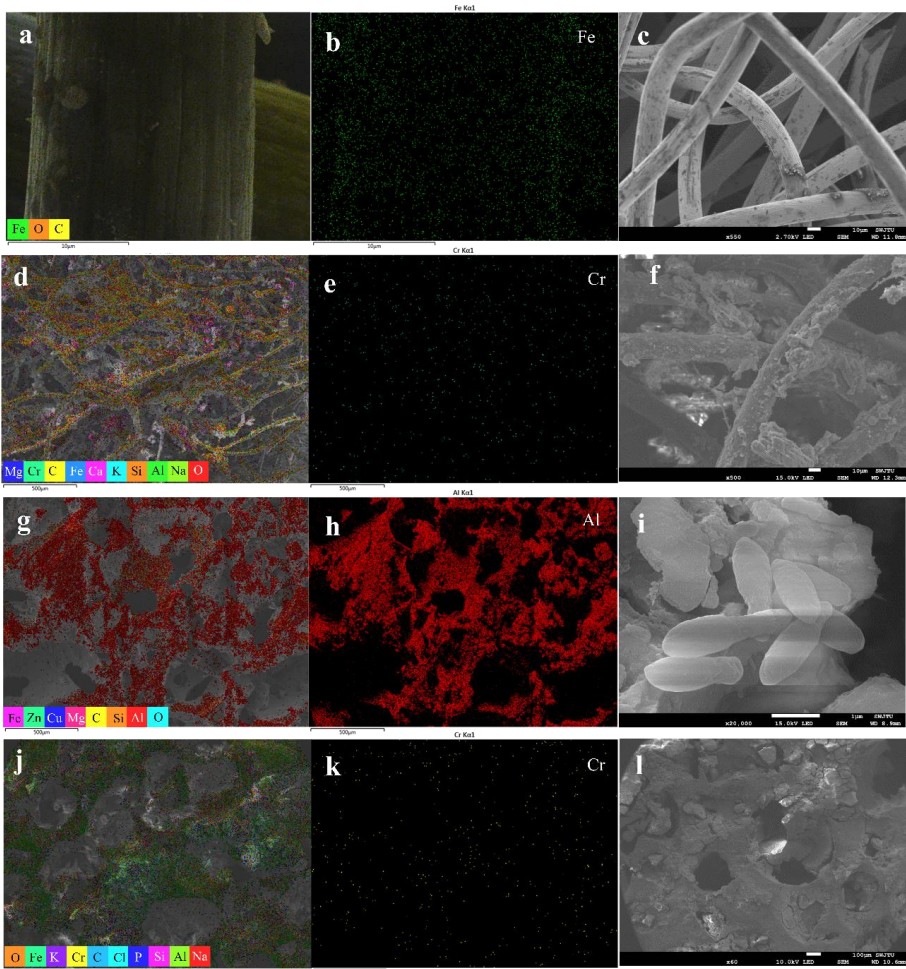

**Fig. 1** Characterization of electrode materials before and after operation by EDS and
SEM. (a-c) EDS and SEM images of cathode loaded with $Fe_3O_4$; (d-f) EDS and SEM
images of cathode after the SMFC operation; (g-h) EDS and SEM images of anode
microorganisms; (j-l) EDS and SEM images of the anode after SMFC operation.

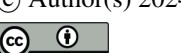


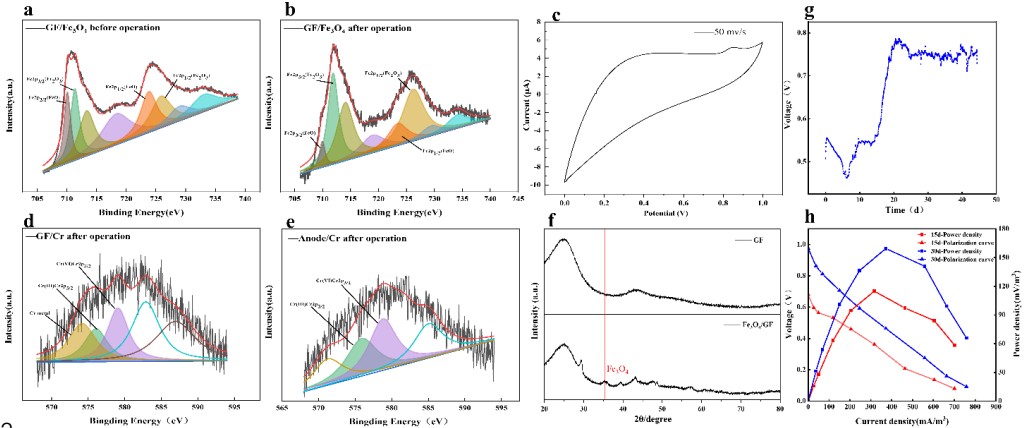

**Fig. 2** Characterization of electrode materials. (a-b) Fe2p spectra of cathode/Fe₃O₄

composite cathode, (c) cyclic voltammetry (CV) curve of cathode/Fe₃O₄, (d-e) Cr2p

spectra of GF composite cathode and Anodic Aluminum foam after operation, (f)

XRD spectrum of the cathode-Fe₃O₄;

Power generation performance of SMFC during long-term operation. (g)output

voltage distribution, (h) 15-day vs. 30-day polarization curves and power density

curves.

**3.2 Electricity Generation Performance**

Initially, CMFC showed a working circuit voltage (WCV) of 0.55 V and an open circuit voltage

(OCV) of 0.68 V (Fig. 2g). In the first week, WCV dropped quickly to 0.45 V but bounced back and

stabilized at 0.75 V on day 25, implying the adaption process of the anode microbes in the soil.

On day 15 (OCV of 0.67 V) and day 30 (OCV of 0.97 V), a series of resistors (50~10 000 Ω) was

connected to the electrodes to determine the polarization curves and power density of the SMFC. As

shown in Fig. 2h, the power density increased and decreased with the elevation of external resistance. At

510 Ω, the power density reached a maximum of 114.9 mW/m³ (73.5 mW/m²) on day 15 and 159.4



mW/m³ (102.0 mW/m²) on day 30 (Table S4). The result indicated that the electrochemical performance
of SMFC enhanced gradually probably due to the microbial adaption. Even after 45 days, the CMFC still
had a WCV of around 0.75 V, indicating its substantial electricity-producing capacity. Compared with
the literature, the SMFC in the current work has an outstanding power generation capacity (Table S5).

**3.3 Cr(VI) reduction and immobilization during operation**

HMs forms determined the bioavailability and toxicity (Jia et al., 2022). After operation, Cr forms
in soils were significantly changed ($P$<0.05) (Fig. 3). In CMFC, the acid-soluble Cr decreased
substantially by 97.44%, the oxidizable and reducible fractions did not change significantly, while the
residual form of Cr increased by 88.89% (Fig. 3a). However, in NMFC, the acid-soluble Cr increased
substantially by 61.54% on day 35. In OMFC, the acid-soluble Cr increased before decreasing, which
was opposite to its oxidizable state. On day 35, the Cr bioavailability of OMFC (11.9%) and NMFC
(18.9%) was 3866-6200 folds of CMFC (0.3%). It is inferred that the electric field and the microbial
communities' evolvement may lead to better Cr immobilization.
In the meantime, the Cr(VI) concentration (Fig. 3c) dropped in all the groups, and Cr (VI) in
CMFC soil was significantly lower than OMFC and NMFC ($P$<0.05) after the experiment. On day 35,
CMFC showed 13.59% and 20.87% higher Cr(VI) elimination than OMFC and NMFC, respectively.
The overlying water was initially free of Cr. During the experiment (Fig. S3), 0.21~12.72 mg/L Cr was
determined, which could be released from the soil. A low level of Cr(VI) (less than 3.15 mg/L) was
detected but vanished later (day 15), which could be attributed to the dynamic adsorption-desorption of
soil particles and electrodes.



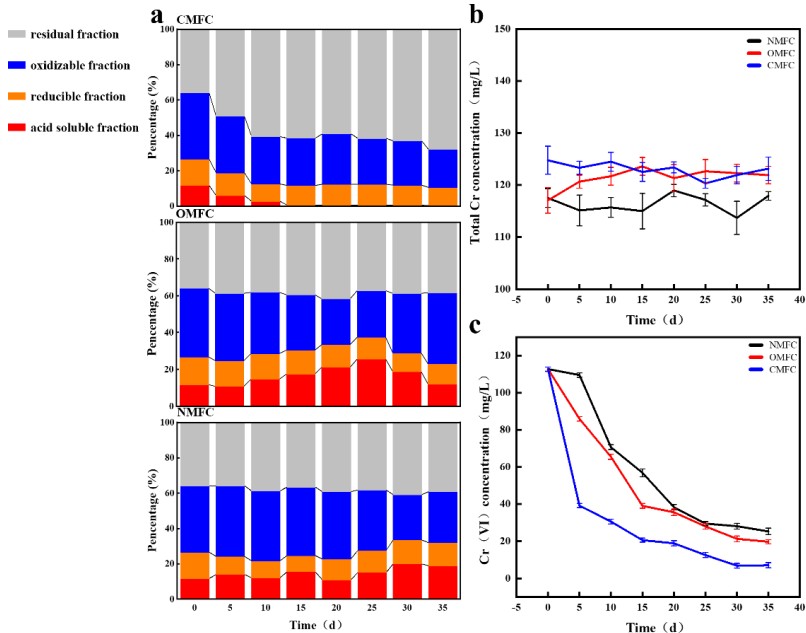

**Fig. 3** (a) Percentage share of Cr in different chemical fractions in CMFC, OMFC, and NMFC soil; Changes in soil (b) total chromium and (c) Cr(VI) concentrations during SMFC operation.

**3.4 Soil properties**

3.4.1 Soil Physicochemical Property

The soil pH decreased in all the groups (Fig. S4A). The pH of CMFC decreased fastest from the initial 7.71 to about 6.83 on day 35, with a minimum of 6.77 on day 30, which was 0.14-7.87% lower than others ($P<0.05$). During the experiment, oxygen in the flooded soil decreased rapidly due to microbial consumption, and acidic products (e.g., low-molecule organic acids) were produced to increase soil acidity. Microorganisms (especially EABs) decompose soil organic matter and release a large



number of electrons and protons, making oxidizing substances such as nitrate and high valence metals
(Fe(Ⅲ), Mn(Ⅳ), and Cr(Ⅵ)) to accept electrons for reduction, causing protons ($H^+$) accumulation (He
et al., 2016). Such a phenomenon was more intense in CMFC due to the rapid electron transfer through
wire to the cathode, leaving protons elevated near the anode.
In all three groups, EC increased rapidly from the initial 1.55 ms/cm before stabilizing (Fig. S4B),
which maximized 2.6, 2.4, and 2.4 ms/cm in CMFC, OMFC, and NMFC ($P$>0.05), respectively. The
rapid increase in EC could be attributed to the inundation that increased the soluble salt content of the
soils. The electromigration in the MFC electric field may also increase soil mass transfer and positively
affect soil electrical conductivity (Zhang et al., 2020).
3.4.2 Soil Biochemical Response
In all the groups, the DHA activity increased significantly (2 244.0~3 138.0% higher than the initial
value) and continuously ($P$<0.05) (Fig. S5A). Under flooding, microbial activity changed from aerobic
to anaerobic, leading to a sharp decline in soil redox potential, accompanied by the stimulation of soil
DHA (Sardans and Peñuelas, 2005). During operation, urease activity in CMFC showed a gradual
increase (2.70~12.40% higher than day 0 from days 10~35), while it in OMFC and NMFC showed a
slight decrease (6.10~7.10% lower than day 0 from days 5~35) (Fig. S5B). SMFC electric field and Fe(II)
promote extracellular electron transfer (EET) (Chen et al., 2023a), which promotes the enrichment of
ammonia-nitrogen transforming bacteria in soil could have caused the higher urease activity in CMFC
than NMFC and OMFC ($P$<0.05). Soil invertase activity decreased initially but increased later for CMFC
and OMFC, but showed an opposite trend for NMFC. After operation, CMFC had a significantly higher
invertase activity than others ($P$<0.05) (Fig. S5C). Soil ACP showed a similar trend with urease, with



CMFC continuously increasing by 13.20~48.90% and considerably higher than OMFC and NMFC (Fig.
S5D).

**3.5 SMFC operation reshaped soil microbial community**

Microbial community structures in the electrodes were analyzed, which obtained 15 dominant phyla
and 50 dominant genera (>1.0%). Overall, *Firmicutes* (73.93%), *Proteobacteria* (62.53%), and
*Chloroflexi* (21.73%) were found the dominant phyla, while *Bacteroides* (21.48%), *Enterococcus*
(17.26%), and *Hyphomicrobium* (15.34%) were the dominant genera.
The alpha diversity analysis indicated a significant difference in the microbial community among
the samples (Fig. 4a). The higher chao1 index in soils than the electrodes demonstrated the higher
microbial richness. Most of the alpha index in OMFC-A and OMFC-C were significantly higher than
CMFC-A and CMFC-C, demonstrating a higher microbial richness and diversity in OMFC. The results
indicated the different microbial evolution patterns in different electrodes and the selection effect of the
electricity field during CMFC operation. The Venn diagram (Fig. S6) found no OTU coincidence among
the samples, indicating their obvious specificity. In comparison, 2130 OTUs were shared by CMFC-S,
OMFC-S, and NMFC-S, indicating the similarity of the soil microbial community (Fig. S6B). 45 OTUs
were shared by Raw-A, CMFC-A, and OMFC-A, accounting for 45.92%, 1.95%, and 1.16%,
respectively, indicating the successful colonization and development of the preloaded EABs.



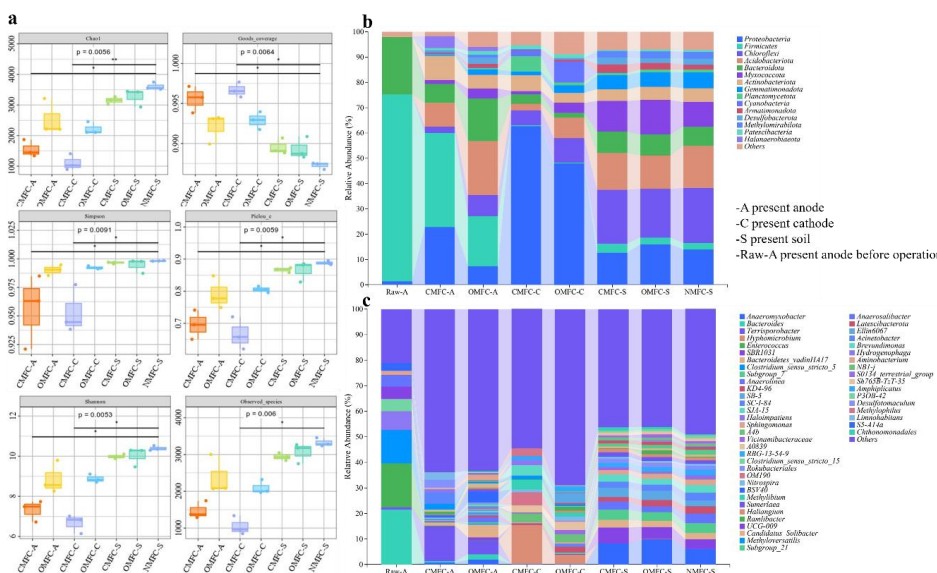

**Fig. 4** (a) Alpha diversity analysis of the electrodes and soils; Microbial community structure is based on (b) the phylum level and (c) the genus level.

3.5.1 Soil microbial community reshaping on phylum level

Before the operation (Fig. 4b), the anode was dominated by *Firmicutes* (73.93%), *Bacteriodota* (22.74%), and *Proteobacteria* (1.27%). After the operation, *Firmicutes* and *Bacteriodota* decreased by 49.83% and 66.84% in CMFC-A, and 73.20% and 26.52% in OMFC-A, respectively. While *Proteobacteria* increased by 1 698.43% in CMFC-A and 475.59% in OMFC-A. Besides, many other phyla emerged, including *Acidobacteriota* (9.41%~21.40%), *Actinobacteriota* (0.23%~9.52%), *Halanaerobiaeota* (1.48%~4.77%), *Myxococcota* (1.51%~3.93%), *Chloroflexi* (0.01%~2.57%), probably due to the penetration of soil indigenous microbe.

The cathode was free of microorganisms initially. However, many phyla were observed after the operation. The CMFC-C was dominated by *Proteobacteria* (62.53%), *Actinobacteriota* (6.24%),



*Planctomycetota* (6.04%), *Chloroflexi* (5.88%), and *Bacteroidota* (3.81%), while OMFC-C was
dominated by *Proteobacteria* (47.93%), *Chloroflexi* (9.75%), *Cyanobacteria* (8.30%), *Acidobacteriota*
(8.01%), *Actinobacteriota* (3.97%), *Myxococcota* (3.92%), and *Gemmatimonadota* (2.87%). The
*Proteobacteria* phylum was rich in EABs, its advantage in both electrodes of CMFC indicated that SMFC
operation was favorable for EAB colonization and development. All the soils were dominated by
*Chlorobacteria*, *Acidobacteria*, *Proteobacteria*, *Bacteroidetes*, and *Myxococcota*, and the difference was
not significant.
3.5.2 Soil microbial community reshaping on genus level
At the genus level (Fig. 4c), MFC operation presented a selection effect, with *Terrisporobacter*
increasing from 0.81% to 13.71% and *Bacteroides* decreasing from 12.48% to 0.53% in CMFC anode.
Compared with the Raw-A, many EABs in CMFC-A decreased, including *Clostridium_sensu_stricto_5*
(from 12.99% to 0.052%), *Clostridium_sensu_stricto_15* (from 4.70% to 0.47%), *Enterococcus* (from
17.26% to 0.03%) (Choi, 2022; Zhang et al., 2023). However, the *Desulfotomaculum* in CMFC-A
increased to 3.32% compared with 0.003% in the soil (CMFC-S). Besides, soil indigenous bacteria
including *Ramlibacter*, *Methyloversatilis,* and *Acinetobacter* colonized in the anode and elevated by
4.89~1 579 fold compared with soil. Nevertheless, multiple dominant genera in the soils decreased in
CMFC-A than in OMFC-A. For example, *SBR1031* accounted for 3.63%~6.18% in the soils, but 0.33%
in CMFC-A and 1.08% in OMFC-A. *Bacteroidetes_vadinHA17* accounted for 2.48%~3.09% in the soils,
but 1.03% in CMFC-A and 5.13% in OMFC-A. *Anaerolinea* accounted for 2.37%~3.63% in the soils,
but 0.25% in OMFC-A, and 1.89% in OMFC-A. The electric field action to a certain extent helped the
anode to resist external microbial intrusion to ensure the stability of the anodic microbial community.



During operation, the prolonged interaction between the soil and water phases resulted in the
gradually evolving unique biofilm structure of the cathode. For instance, *Hyphomicrobium* (3.56~15.34%
in soils), an aerobic chemoheterotroph capable of degrading a wide range of organics, accounted for
15.34% and 3.56% of CMFC-C and OMFC-C, respectively (He et al., 2019). *Hydrogenophaga*, a gram-
negative bacteria capable of denitrification and Cr(VI) reduction, accounted for 2.07% of CMFC-C
(Wang et al., 2022). Meanwhile, the SMFC operation caused the enrichment of several resistant bacteria.
*Subgroup_7*, a typical HM-tolerant bacterium (Li et al., 2023a), was enriched in both cathode and soil.
*Acinetobacter* and *Limnohabitans*, also tolerators that carry HRGs and ARGs, were found 4.31% and
3.03% in CMFC-A (Dahal et al., 2023; Zhang et al., 2021).
The increase of iron in the soil and water due to the use of $Fe_3O_4$ as the cathode catalyst may be
responsible for the enrichment of *Terrisporobacter* and *Anaeromyxobacter* in the CMFC-A and OMFC-
A. They were found closely associated with $Fe^{3+}$ reduction to gain energy in various environments (Lin
et al., 2007; Wang et al., 2020b).
3.5.3 Soil metal resistance gene variation
Under Cr(VI) stress, certain microbes would utilize pathways like specific or non-specific Cr(VI)
reduction, free radical detoxification, DNA damage repair, etc. to survive in toxic environments (Morais
et al., 2011). Using qPCR analysis, the abundance of typical HRGs and MGEs in the anodic soils was
determined (Fig. 5a), which varied greatly during operation ($P<0.05$). Compared with OMFC and NMFC,
*chrA* in CMFC increased by 237.83% and 3414.34%, *chrB* by 141.52% and 153.63%, *chrR* by 221.86%
and 839.41%, *IntI* by 151.77% and 167.91%, *tnpA02* by 331.86% and 1118.97%, and *tnpA05* by 416.91%
and 99.54%.



The elevation of HRGs and MGEs could be due to the enrichment of multiple metal-resistant
bacteria (MRB) such as *Acinetobacter, Limnohabitans,* and *Brevundimonas*. Moreover, the anodic
*Desulfotomaculum*, which accounted for 3.23% of CMFC-A, is a typical sulfate-reducing bacterium
(SRB) that produces $H_2S$, a natural signaling molecule that contributes to tolerance triggering,
maintenance, and diffusion through community sensing, which facilitates HRG elevation through HGT
(Shatalin et al., 2021). Besides, Cr(VI) reducing bacteria like *Hydrogenophaga* (1.31% in CMFC-A) may
also up-regulate the Cr reductase gene *chrR* (Sundarraj et al., 2023).
Furthermore, pH changes may also affect soil resistance characteristics. Liu et al. (2023c) observed
the abundance of multidrug efflux pump genes in the acid soil was significantly positively correlated
with soil acidity. The intensified proton generation and accumulation in CMFC could have led to the
HRG elevation. In addition, HMs toxicity exerts direct selective pressure, which affects microbial
community structure and their function, leading to the thriving of tolerators like *Desulfotomaculum sp*,
*Hydrogenophaga*, and *Methylophilus* (Hernández-Ramírez et al., 2018), hence the spontaneous HRG
elevation(Wang et al., 2023a).



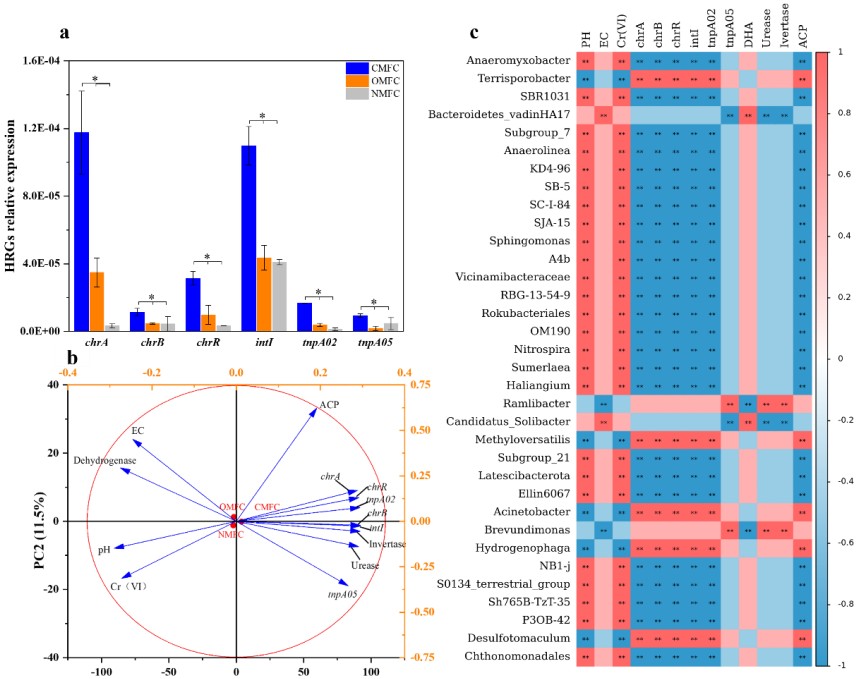

**Fig. 5** (a)qPCR results of heavy metal resistance gene changes in soil around anode-

soil after SMFC operation (*chrA*, *chrB*, *chrR*, *intI*, *tnpA02*, *tnpA05*). Different letters

denote significant differences among treatments (P < 0.05) **;** (b) principal component

analysis (PCA) of SMFC soil physicochemical properties with Enzyme activity,

HRGs, and native bacterial genera and (c) Spearman's correlation heatmap (mean

relative abundance > 1%). * P < 0.05, according to LSD test (mean ± S.E., n = 3);* *

P < 0.01.

## 3.6 Correlation analysis

To visually analyze the correlation between bacterial communities and environmental factors,



spearman correlation analysis and principal component analysis (PCA) were conducted (Fig. 5b).
Spearman correlation analysis (Fig. 5c) isolated four main bacterial genera clusters. Cluster 1
(*Terrisporobacter, Methyloversatilis, Acinetobacter, Hydrogenophaga, and Desulfotomaculum*) was
positively correlated with HRGs (*chrA, chrB, intI, tnpA02*) and ACP ($P<0.01$), but negatively correlated
with pH and Cr(VI) ($P<0.01$), suggesting they may contribute to HGT and HRGs enrichment. Cluster 2
(*Anaeromyxobacter, Subgroup_7, Anaerolinea, SB-5, Sphingomonas,* etc.) was negatively correlated
($P<0.01$) with HRGs (*chrA, chrB, intI, tnpA02*) and ACP, but positively correlated with pH and Cr(VI)
($P<0.01$). Cluster 3 (*Bacteroidetes_vadinHA17* and *Candidatus_Solibacter*) was positively correlated
($P<0.01$) with soil EC and DHA. Cluster 4 (*Ramlibacter*, and *Brevundimonas*) were positively correlated
with urease and invertase but negatively correlated with EC.

PCA analysis (Fig. 5b) found that soil pH, EC, Cr(VI) concentration, and DHA were positively

correlated with each other but negatively correlated with urease, ACP, invertase, and HRGs. Especially,
Cr(VI) was significantly negatively correlated with HRGs ($P<0.01$), which could be partially explained
by the tolerators thriving and Cr(VI) reduction during SMFC operation.
**4.   Discussion**

In this study, Cr(VI) reduction, microbial community variation, and HRG fate in SMFC were

investigated for the first time. The results proved that SMFC was an effective method to eliminate Cr(VI)
(93.76%), immobilize Cr (97.44%), and generate electricity (0.97 V).

In the SMFC system, Cr(VI) reduction was a synergic result of adsorption/biosorption,

bioelectrochemistry reduction, and microbial reduction (Fig. 6). The preloading of $Fe_3O_4$ and EABs on
the electrodes significantly improved Cr(VI) reduction and power generation by accelerating SMFC



stabilization. The Cr forms, soil physicochemical properties, soil enzyme activities, and microecological
structure mirrored each other, helping to understand Cr transformation patterns and target the key factors
affecting metal resistance changes. The detailed explanation is as follows:
(1) The electricity-producing process of SMFC can inhibit HMs' release and migration in soil (Feng
et al., 2024; Zhu et al., 2019). In this study, Cr forms changed greatly from acid-soluble to a more stable
residual fraction. The $Fe_3O_4$-modified cathode not only directly adsorbs or reduces Cr(VI) due to the high
specific area and ferrous iron, but also enhances the electrochemical effect of the system. The electrons
derived from anodic microbial metabolism can directly reduce Cr(VI) in the soil to Cr(III), while part of
them is transmitted to the cathode, where Cr(VI) in the overlying water compete with oxygen as electron
acceptors and complete the current loop (Thapa et al., 2022).
(2) Microorganisms can also directly or indirectly reduce or fix Cr. Biosorption, sulfide, and
hydroxide precipitation are the main immobilization mechanisms of HMs by microorganisms (Ma et al.,
2024). For example, *Desulfotomaculum sp.*, a typical SRB, enriched to 3.23% in CMFC-A, may produce
sulfide ions by reducing alienated sulfate, thus forming highly insoluble metal sulfide to fix Cr through
microorganism-induced sulfide precipitation (MISP). *Hydrogenophaga*, which dominated in both
electrodes, was a known Cr-reducing bacteria. Some iron-reducing bacteria (e.g., *Anaeromyxobacter* and
*Terrisporobacter*) may also contribute to Cr(VI) reduction by participating in the Fe cycle through EET,
while the ferrous iron reduces Cr(VI). Additionally, the CMFC in this work contains many genera capable
of transforming nitrogen. For example, *Hyphomicrobium*, a typical denitrifying bacterium, that can
effectively reduce nitrate and nitrite (Ernst et al., 2021), dominated in CMFC-C (15.34%). *Methylophilus*,
a methylotrophic microorganism (Yang et al., 2020), accounted for 2.93% of CMFC-A but was much





lower in other groups. The bacteria mentioned above were found with high urease-producing ability,
whose enrichment not only improves soil urease activity and nutrient cycling but also immobilizes Cr
through microorganism-induced carbonate precipitation (MICP) (Qian et al., 2017).

(3) Under electrochemical selection and HMs stress, the microbial community gradually evolved

with higher richness and diversity, along with the HRG enrichment and nutrient cycling vriation. Firstly,
some soil indigenous bacteria were much lower in CMFC-A than OMFC-A, indicating the electric field
contributed to the anode stability by preventing external bacteria intrusion and is less vulnerable to
environmental fluctuations. The microbial community change is significantly related to HRG enrichment.
Many EABs and MRBs are significantly enriched. For example, *Desulfotomaculum*, an SRB with a dual
role of electroproduction and HMs reduction (Jiang et al., 2020; Yin et al., 2021). Other examples also
include cumulative-resistant bacteria like *Acinetobacter* and *Limnohabitans* that are only enriched in the
CMFC-A (AL-Jabri et al., 2018; Dahal et al., 2023). Their enrichment directly causes vertical gene
transfer and HRG elevation.

HMs existence in soil can induce HGT occurrence and cause ARG elevation, which has become a

major concern (Chen et al., 2023b; Fu et al., 2023). Sub-lethal levels of metal ions can increase mutation
rates and enrich de novo mutants with significant resistance to multiple antibiotics (Li et al., 2019). This
study focused on the toxic alleviation of a single HM (Cr) in SMFC, during which tolerator accumulation
caused considerable HRG enrichment. SMFC is an eco-friendly and cost-effective technology for the in-
situ bioremediation of contaminated soil/sediment and powering environmental sensors in remote areas.
It has the potential to be used as a novel early warning system for soil environmental hazards.
Nevertheless, before the commercialization of large-scale applications in the field, significant efforts

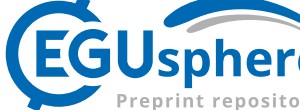

should be made to reveal the HRG enrichment mechanism during SMFC operation and pay attention to
ARG change under HMs contamination or HMs-antibiotic co-contamination.

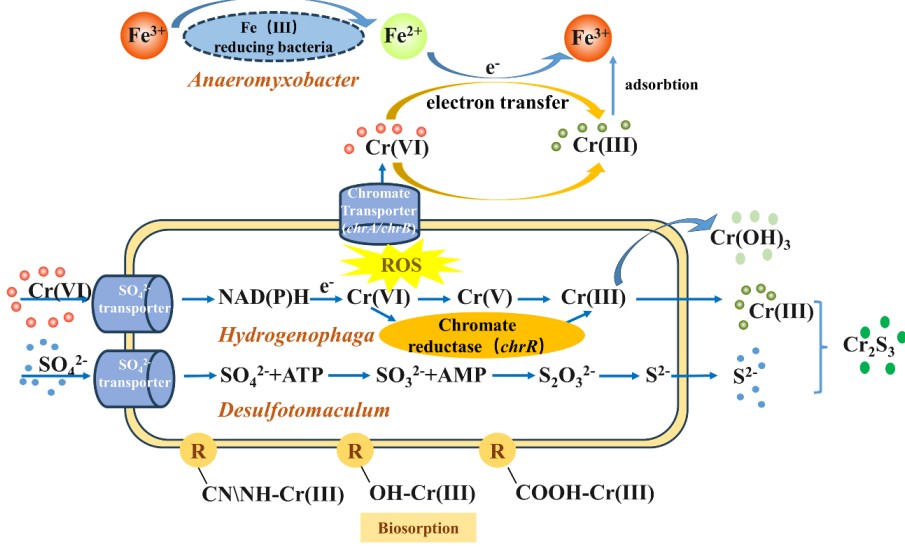

**Fig. 6** Cr(VI)reduction Mechanism of during SMFC operation

**5.   Conclusion**
During SMFC operation, soil physicochemical properties, enzyme activities, resistance genes, and
microbial community structure closely interacted with each other. The pre-loading of $Fe_3O_4$ in the
cathode and EAB in the anode greatly contributed to the power production and Cr(VI) elimination.
Anodic microbial metabolism, cathodic redox, and the MFC electric field reduced or immobilized Cr(VI)
to eliminate its risk. The enrichment of multiple MRBs, such as *Acinetobacter, Limnohabitans,* and
*Desulfotomaculum*, resulted in HRG elevation, which contributes to microbial adaptation and function
but brings concerns for future application. This study provides a reference for the remediation of HM-
contaminated soil using MFC, which is conducive to promoting the practical application of





bioelectrochemical technology in the field.
**Author contribution:**
Huan Niu: Conceptualization, Investigation, and Writing; Xia Luo: Investigation, Visualization;
Peihan Li: Investigation, Visualization; Haitao Ma: Methodology; Hang Qiu: Methodology; Liyue Jiang:
Writing-review and editing; Subati Maimaitiaili: Writing-review and editing; Minghui Wu: Funding
acquisition, Supervision; Fei Xu: Funding acquisition, Supervision; Heng Xu: Funding acquisition,
Supervision; Can Wang: Funding acquisition, Supervision.
**Acknowledgments**
This work was financially supported by the Sichuan Science and Technology Program
(2024NSFSC0384), the Government Guides Local Science and Technology Development Project of
Tibetan Autonomous Region of China (XZ202401YD0001), the China Postdoctoral Science Foundation
(2022M712630), and the Fundamental Research Funds for the Central Universities (2682023ZTPY079).
We thank Dr. Weizhen Fang from the Analysis & Testing Center and Dr. Cuijuan Wang from the School
of Chemistry, Southwest Jiaotong University, for the technical support. We thank Lijuan Zhang from
SCI-Go (www.sci-go.com) for the XPS analysis.
**Declaration of interests**
The authors declare that they have no known competing financial interests or personal relationships
that could have appeared to influence the work reported in this paper.

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
