# Peer review of "Cr(VI) reduction, electricity production, and microbial resistance variation in"

_EGUsphere, 2024_

## Author Comment (AC1)

7. **The Cr shown in both Figure 1e and Figure 1k is easy to be confused. It is recommended to mark them in the appropriate position in the diagram.**

**Response:** Thank you for your kind suggestion. We will attach clear and eye-catching ICONS to the relevant images in the revised manuscript.

[Figure]

**Fig. 1** Characterization of electrode materials before and after operation by EDS and SEM. (a-c)

EDS and SEM images of cathode loaded with $Fe_3O_4$; (d-f) EDS and SEM images of cathode after

the SMFC operation; (g-h) EDS and SEM images of anode microorganisms; (j-l) EDS and SEM

images of the anode after SMFC operation.

8. **Figure 2 shows that the cathode is GF while the anode is not represented by the material name. Please uniformly use the electrode or material noun to provide a good reading experience.**

**Response:** Thanks for the suggestion. We will use uniform material nouns in the revised manuscript. And Figure is replotted as follows:

[Figure]

**Fig. 2** Characterization of electrode materials. (a-b) Fe2p spectra of cathode/Fe₃O₄ composite

cathode, (c) cyclic voltammetry (CV) curve of cathode/Fe₃O₄, (d-e) Cr2p spectra of GF composite

cathode and Anodic Aluminum foam after operation, (f) XRD spectrum of the cathode-Fe₃O₄,

Power generation performance of SMFC during long-term operation. (g)output voltage

distribution, (h) polarization curves and power density curves (15-day vs. 30-day).

---

## Author Comment (AC2)

**Review of "Cr(VI) reduction, electricity production, and microbial resistance variation in paddy soil under microbial fuel cell operation" by Niu et al. for consideration in EGUsphere.**

**General comments**

**Niu et al. constructed a soil MFC to reduce Cr(VI) in paddy soil and investigate its impact on the microbial community and microbial resistance traits. The findings provide some important clues supporting the application of bioelectrochemical systems in contaminated paddy soils, offering a potential approach for environmental remediation and detoxification. However, several details require further improvement before being considered for publication.**

We thank the reviewers for their time and feedback on this manuscript. Our point-to-point responses are below.

**Specific comments**

1. **Please explain why aluminum foam was chosen over other materials and what advantages it offers in this experiment.**

**Response:** Thank you for your suggestion.

In this experiment, the anode is operated in a low oxygen moist soil environment, and the soil medium composition is complex, requiring the anode to have high corrosion resistance. Porous metal skeletons, such as foam copper, foam nickel and foam aluminum, are known for their good conductivity and ability to evenly distribute the metal-ion flux through spatial segmentation[2]. Among them, foam aluminum is highly potential as an electrochemical substrate material due to its light weight, low cost and high abundance[1][3]. The material has a very low density, good electrical conductivity, and resistance to acid and alkali corrosion[4]. In addition, aluminum foam's unique multi-space structure not only helps load more functional bacteria in the early stage but also increases the anode conducti on electron efficiency of SMFC with its excellent specific surface area.

Due to the good corrosion resistance, electricity conductivity, and high specific surface area, aluminum foam was chosen as the anode material.

Some references are as follows:

[1] Chen, J., Wang, Y., Li, S., Chen, H., Qiao, X., Zhao, J., Ma, Y., Alshareef, H.N., (2023). Porous Metal Current Collectors for Alkali Metal Batteries. Advanced Science 10(1), 2205695. https://doi.org/https://doi.org/10.1002/advs.202205695.

[2] Ding, Y., Zhang, Q., Rui, K., Xu, F., Lin, H., Yan, Y., Li, H., Zhu, J., Huang, W., (2020). Ultrafast Microwave Activating Polarized Electron for Scalable Porous Al toward High-Energy-Density Batteries. Nano Letters 20(12), 8818-8824. https://doi.org/10.1021/acs.nanolett.0c03762.

[3] Lamiel, C., Hussain, I., Ma, X., Zhang, K., (2022). Properties, functions, and challenges: current collectors. Materials Today Chemistry 26, 101152. https://doi.org/https://doi.org/10.1016/j.mtchem.2022.101152.

[4] Rossi, S., Bizzotto, M., Deflorian, F., Fedel, M., (2019). Study of anodizing process on aluminium foam to improve the corrosion behavior. Surface and Interface Analysis 51(12), 1194-1206. https://doi.org/https://doi.org/10.1002/sia.6610.

2. **In the manuscript, the authors included a virtual model diagram of the SMFC. Please include a realistic photograph of the setup in the subsequent revision to enhance clarity and provide a more comprehensive understanding.**

**Response:** Thanks for the suggestion. We will add a realistic photograph of the SMFC device to the supplementary material. The picture is as follows:

The left part of the picture is the SMFC photo, and the right part is the model groups. A plastic box (140.0×85.0×165.0 mm) was used as the SMFC reactor, with 1.50 kg soil and overlying water of 3.0 cm to simulate the flooded state during rice planting. The cathode was floated on the water surface while the anode was buried (about 3.0 cm from the bottom). The cathode and anode were connected to a 2000 Ω resistor using titanium wire. The water level was kept constant by daily replenishment.

[Figure]

**Fig. S1** SMFC structure and experimental grouping.

3. **In the abstract, the authors state: 'Fe₃O₄ nanoparticle as the cathodic catalyst effectively boosted power generation (0.97 V, 102.0 mW/m²), whose porous structure and reducibility also contributed to Cr reduction and immobilization.' However, there is a lack of introduction regarding the cathode catalysts.**

**Response:** Thank you for your professional advice. In the revised manuscript, the related paragraph was revised as:

Graphite felt (GF) is fabricated from the polyacrylonitrile through a wiredrawing, high-temperature carbonization and graphitization process, in which a trapezoidal hexagonal ring-like lamellar structure is formed in the microfibrillar of the polyacrylonitrile-based carbon fiber. GF is widely used as a cathode due to its non-toxicity, stability, good electrical conductivity, chemical resistance. Moreover, it has a large three-dimensional active surface and mechanical integrity [1][2].

Some of the newly added references are as follows:

[1] Miao, J., Zhu, H., Tang, Y., Chen, Y., Wan, P., (2014). Graphite felt electrochemically modified in H2SO4 solution used as a cathode to produce H2O2 for pre-oxidation of drinking water. Chemical Engineering Journal 250, 312-318.

[2] Zhao, K., Quan, X., Chen, S., Yu, H., Zhang, Y., Zhao, H., (2018). Enhanced electro-Fenton performance by fluorine-doped porous carbon for removal of organic pollutants in wastewater. Chemical Engineering Journal 354, 606-615.

4. **The manuscript mentions the source and size of anode and cathode materials in section 2.2.2 Electrodes Preparation. However, some essential characteristics, such as the porosity and bulk density of the materials, are missing. Including these details is necessary to provide a comprehensive understanding of the electrode properties.**

**Response:** Thank you for your professional advice. In the revised manuscript, the related paragraph was revised as:

Aluminum foam ($66.0 \times 54.0 \times 5.0$ mm, porosity 60-80%, bulk density 0.50-1.10 g/cm$^3$) (SANZHENG Metal material, Chengdu, China) was used as anode. The anode microflora was derived from municipal sludge (Chengdu Sixth Sewage Treatment Plant, China) after acclimating with 100 mg/L Cr(VI). Before assembling, the aluminum foam was cultivated in the anode microflora for 2 weeks. Then the anode was tied to titanium mesh tightly with titanium wire. Graphite felt (GF) ($100.0 \times 50.0 \times 3.0$ mm, bulk density 0.10-0.15 g/cm$^3$) was used as the cathode (Table S1). Before use, it was cleaned, dried, and loaded with $Fe_3O_4$ as the ORR catalyst, as detailed in section 1 of the supplementary material. For characterization, we utilized a scanning electron microscope (SEM) to examine the structure and morphology of the electrode surface. In addition, we performed X-ray photoelectron spectroscopy (XPS) and energy dispersive spectroscopy (EDS) to analyze the valence state and element composition. The phase composition was determined using an X-ray diffractometer (XRD).

5. **Lines 138-143: Please provide detailed information on HRGs determination, including the reagents, instruments, and procedures used.**

**Response:** Thank you for your professional advice. In the revised manuscript, the related paragraph was revised as:

Total bacterial DNA was extracted using an E.Z.N.A.® Soil DNA Kit (Omega Biotek Inc., USA) according to the manufacturer's protocol. Specifically, 0.50 g sample, 0.50 g magnetic beads, and 1.0 ml SLX-Mlus Buffer were added in a 2.0 ml Eppendorf tube, and ground for 250 s under 45 HZ. Then added and mixed with 100 µl DS Buffer, and cultivated under 70 ℃ for 10 min and then 90℃ for 2 min. Then the mixture was centrifuged at 10000 g for 5 min under room temperature. 800 µl supernatant was moved to a new tube and added with 270 µl P2 buffer and 100 µl HTR reagent, and then cultivated under -20 ℃ for 5 min and then centrifuged again at 10000 g for 5 min. The supernatant was then moved to a new 2 ml tube and added with the same amount of XP5 buffer and mixed upside down for 8 min. After magnetic rack adsorption, discard the residual liquid, remove the tube, add 500 µL XP5 Buffer, and mix well. Then adsorbed again with a magnetic rack, discard the residual liquid, remove the tube, add 500 µL PHB, and mix well. Then adsorbed again with a magnetic rack, discard the residual liquid, remove the tube, add 500 µL SPW Wash Buffer, and mix well (repeat this step twice). Then the mixture was adsorbed again with a magnetic rack, discard the residual liquid, centrifuge the tube under 10000 g for 10 s. Then the beads were adsorbed again with a magnetic rack, discard the residual liquid, and let stand for 8 min. After that, the beads were added with 100 µL elution buffer, mixed, and let stand for 5 min. Finally, after adsorbing with a magnetic rack, the supernatant was moved to a new 1.0 ml Eppendorf tube, and total DNA was obtained for further use. The PCR reaction system was constructed. The abundance of HRGs in the surface soil of SMFC and OMFC anode after operation was analyzed using an SYBR Green real-time fluorescence quantitative PCR system (7500, Thermo Fisher, USA) (Wang et al., 2023a). The

soil of OMFC was used for comparison. The detected genes included HRGs (*chrA, chrB, chrR, recG, nfsA, zupT, fpvA*) and MGEs (*intI, tnpA02, tnpA04, tnpA05*). The primer sequences are provided in Table S2.

**Technical corrections**

1. **Please check typos throughout the text. As an example, supplementary material Table S5, 'conFig.urations'?**

**Response:** Thank you for your careful examination. During revision, we will carefully review and revise the typos to ensure its accuracy.

 The "conFig.urations " will be amended to "configurations ".

2. **Some figures in the manuscript need to be improved.**
**Line 167: The annotations in Figure 1 are too small to be clearly seen.**
**Line 172: the icon in Figure 2 is blurred, and the the resolution need to be improved.**

**Response:** Thank you for your kind suggestion. We will attach clear and eye-catching ICONS to the relevant images in the revised manuscript.

[Figure]

**Fig. 1** Characterization of electrode materials before and after operation by EDS and SEM. (a-c) EDS and SEM images of cathode loaded with $Fe_3O_4$; (d-f) EDS and SEM images of cathode after the SMFC operation; (g-h) EDS and SEM images of anode microorganisms; (j-l) EDS and SEM images of the anode after SMFC operation.

[Figure]

**Fig. 2** Characterization of electrode materials. (a-b) Fe2p spectra of cathode/Fe₃O₄ composite cathode, (c) cyclic voltammetry (CV) curve of cathode/Fe₃O₄, (d-e) Cr2p spectra of GF composite cathode and Anodic Aluminum foam after operation, (f) XRD spectrum of the cathode-Fe₃O₄, Power generation performance of SMFC during long-term operation. (g)output voltage distribution, (h) polarization curves and power density curves (15-day vs. 30-day).

---

## Author Comment (AC4)

**Review of "Cr(VI) reduction, electricity production, and microbial resistance variation in paddy soil under microbial fuel cell operation" by Niu et al. for consideration in EGUsphere.**

**The manuscript entitled " Cr(VI) reduction, electricity production, and microbial resistance variation in paddy soil under microbial fuel cell operation " has been written well and supported with sufficient experimental data. The authors found that during SMFC operation, soil physicochemical properties, enzyme activities, resistance genes, and microbial community structure closely interacted with each other. Although, the overall design of experiment and manuscript are fine, but some changes are required to be made in manuscript. Therefore, following changes should be included before publication:**

We would like to thank the reviewers for their time and feedback on this manuscript. Please find our point-to-point responses below.

1. **Whether the SMFC has application prospects in practice, whether it can be applied to other soils except rice field soil, and whether the $Fe_3O_4$ will cause secondary pollution to the environment. Its advantage as cathode catalyst should also be mentioned. Please give a brief description.**

**Response:** Thanks for the suggestion.

This study presents an innovative treatment strategy that is economical and eco-friendly compared with conventional physicochemical methods. The soil MFC (SMFC) configuration was designed for in situ treatment and allows the anode to utilize the organic contents in sediment without a separator to divide the anode and cathode. Moreover, nutrients are passively supplemented during sedimentation from contaminated water to provide sustainable electron donors for SMFCs. The method can be used for in-situ contaminant treatment in a variety of environments, such as aquaculture ponds, inland lakes, and wetlands.

The iron oxides ($Fe_3O_4/Fe_2O_3$) are most commonly used as the modifier. The iron oxides modification mainly improves the kinetics activity of the reaction[1]. Iron-based materials due to the formation of dense flocs with good settling properties that remove phosphorus and other pollutants by adsorption and sweeping, have been widely used in environmental treatment[3]. Iron is a critical component of cytochrome C and iron-sulfur proteins, both of which are required by most electricigens and play a crucial role in the respiratory chain of microorganisms [2]. Therefore, the appropriate amount of $Fe_3O_4$ residue in the soil will not cause secondary pollution.

Some references are as follows :

[1] Li, J., Gao, H., (2008). A Renewable Potentiometric Immunosensor Based on Fe3O4 Nanoparticles Immobilized Anti-IgG. Electroanalysis 20(8), 881-887. https://doi.org/https://doi.org/10.1002/elan.200704094.

[2] Yu, B., Li, Y., Feng, L., (2019). Enhancing the performance of soil microbial fuel cells by using a bentonite-Fe and Fe3O4 modified anode. Journal of Hazardous Materials 377, 70-77. https://doi.org/https://doi.org/10.1016/j.jhazmat.2019.05.052.

[3] Venkateswarlu, S., Yoon, M., Kim, M.J., (2022). An environmentally benign synthesis of Fe3O4 nanoparticles to Fe3O4 nanoclusters: Rapid separation and removal of Hg(II) from an aqueous medium. Chemosphere 286, 131673. https://doi.org/https://doi.org/10.1016/j.chemosphere.2021.131673

**2. Some of the formatting of the manuscript needs to be improved for ease of reading. Iines 278-290, the paragraph is too long to grasp the point and requires the author to break it up in appropriate places.**

**Response:** Thank you for your professional advice. In the revised manuscript, the related paragraph was revised as:

At the genus level (Fig. 4c), MFC operation presented a selection effect, with *Terrisporobacter* increasing from 0.81% to 13.71% and *Bacteroides* decreasing from 12.48% to 0.53% in CMFC anode. Compared with the Raw-A, many EABs in CMFC-A decreased, including *Clostridium_sensu_ stricto_5* (from 12.99% to 0.052%), *Clostridium_sensu_stricto_15* (from 4.70% to 0.47%), and *Enterococcus* (from 17.26% to 0.03%) (Choi, 2022; Zhang et al., 2023).

However, the *Desulfotomaculum* in CMFC-A increased to 3.32% compared with 0.003% in the soil (CMFC-S). Besides, soil indigenous bacteria including *Ramlibacter, Methyloversatilis*, and *Acinetobacter* colonized in the anode and elevated by 4.89~1 579 fold compared with soil. Nevertheless, multiple dominant genera in the soils decreased in CMFC-A than in OMFC-A. For example, *SBR1031*, *Bacteroidetes_vadinHA17*, and *Anaerolinea* were significantly increased in soil, but less in CMFC-A and OMFC-A. The electric field action to a certain extent helped the anode to resist external microbial intrusion to ensure the stability of the anodic microbial community.

**3. Please explain the sampling process during the SMFC operation and if it will affect the normal operation and cause large errors?**

**Response:** Thank you for your professional advice.

We used a plastic cylindrical straw with a diameter of 0.4 cm and a length of 16 cm as the sediment sampler. The SMFC sediment part is inserted by the sampler vertically at a specific time, and then quickly removed, and the upper, middle and lower parts of the sampler are mixed as a determination sample. And the fresh sample each time is only 8-16 g, only 0.2-0.4% of the SMFC. The total sampling amount shall not exceed 5% of the total population. Because the sampler is much smaller than SMFC, the disturbance is avoided to a great extent and the normal operation of SMFC is guaranteed.

**4. Don't use the notion like 'we' or 'our' etc., as these are the redundant words (not the research words) for the standard journal manuscripts.**

**Response:** Thank you for your kind advice. According to the reviewer's suggestion, the manuscript was carefully revised to make sure the use of research words, and some unnecessary redundant words were removed.

**5. In part 2.5.3 HRG Fluctuation, does the author refer to extracted DNA for HRG detection? Please add the specific steps of extraction.**

**Response:** Thank you for your kind advice. According to the reviewer's suggestion, the details of DNA extraction from the sample were added and marked in red in the material and method section. The detailed DNA extraction procedure was as follows:

Total bacterial DNA was extracted using an E.Z.N.A.® Soil DNA Kit (Omega Biotek Inc., USA) according to the manufacturer's protocol. Specifically, 0.50 g sample, 0.50 g magnetic beads, and 1.0 ml SLX-Mlus Buffer were added in a 2.0 ml Eppendorf tube, and ground for 250 s under 45 HZ. Then added and mixed with 100 µl DS Buffer, and cultivated under 70 ℃ for 10 min and then 90℃ for 2 min. Then the mixture was centrifuged at 10000 g for 5 min under room temperature. 800 µl supernatant was moved to a new tube and added with 270 µl P2 buffer and 100 µl HTR reagent, and then cultivated under -20 ℃ for 5 min and then centrifuged again at 10000 g for 5 min. The supernatant was then moved to a new 2 ml tube and added with the same amount of XP5 buffer and mixed upside down for 8 min. After magnetic rack adsorption, discard the residual liquid, remove the tube, add 500 µL XP5 Buffer, and mix well. Then adsorbed again with a magnetic rack, discard the residual liquid, remove the tube, add 500 µL PHB, and mix well. Then adsorbed again with a magnetic rack, discard the residual liquid, remove the tube, add 500 µL SPW Wash Buffer, and mix well (repeat this step twice). Then the mixture was adsorbed again with a magnetic rack, discard the residual liquid, centrifuge the tube under 10000 g for 10 s. Then the beads were adsorbed again with a magnetic rack, discard the residual liquid, and let stand for 8 min. After that, the beads were added with 100 µL elution buffer, mixed, and let stand for 5 min. Finally, after adsorbing with a magnetic rack, the supernatant was moved to a new 1.0 ml Eppendorf tube, and total DNA was obtained for further use.

**6. For gene extraction was it dried or fresh samples?**

**Response:** Thank you for your careful work. For DNA extraction, 0.50 g fresh sample was used after homogenization. The detailed information was added and marked in red in the material and method section of the revised manuscript.

The related paragraph was revised as:

Total bacterial DNA was extracted using an E.Z.N.A.® Soil DNA Kit (Omega Biotek Inc., USA) according to the manufacturer's protocol. Specifically, 0.50 g fresh sample, 0.50 g magnetic beads, and 1.0 ml SLX-Mlus Buffer were added in a 2.0 ml Eppendorf tube, and ground for 250 s under 45 HZ.

**7. Lines 157-159, " After operation, the typical peaks of Cr(Ⅲ), Cr(Ⅵ), and element Cr (576.1 and 578.92) were observed on both electrodes by XPS (Fig. 2d-e) " are insufficient to explain that the reduction and immobilization of Cr(VI) by the electrodes.**

**Response:** Thank you for your careful work.

Chromium exists mainly in the III and VI oxidation states in soil. The peaks at 576.9eV and 579eV are typical peaks of Cr(Ⅲ) and Cr(Ⅵ), respectively. After operation, the typical peaks of

Cr(Ⅲ) and Cr(Ⅵ) can be observed on the X-ray energy spectra of both cathode and anode, and Cr(Ⅵ) and Cr(Ⅲ) are present on the cathode and cathode of SMFC. The results showed that at least the valence state of Cr changed on the electrode and Cr(VI) was reduced and fixed by the electrode[1].

Some references are as follows:

[1]  Kim, C., Lee, C.R., Song, Y.E., Heo, J., Choi, S.M., Lim, D.-H., Cho, J., Park, C., Jang, M., Kim, J.R., (2017). Hexavalent chromium as a cathodic electron acceptor in a bipolar membrane microbial fuel cell with the simultaneous treatment of electroplating wastewater. Chemical Engineering Journal 328, 703-707. https://doi.org/https://doi.org/10.1016/j.cej.2017.07.077.

---

## Author Response (AR2)

**Cr(VI) reduction, electricity production, and microbial resistance variation in paddy soil under microbial fuel cell operation**

**Manuscript Number: egusphere-2024-2771R1**

Dear Editor,

Thank you for your letter and the comments concerning our manuscript. Those comments are valuable and helpful for improving our paper. We have studied these comments carefully and have made the corresponding corrections. We hope the revised manuscript will meet your approval.

The response to the editor and reviewer's comments is as follows.

**Responds to Comments**

**Reviewer #1:**

**The manuscript titled " Cr(VI) reduction, electricity production, and microbial resistance variation in paddy soil under microbial fuel cell operation " (egusphere-2024-2771) focused on SMFC for Cr(VI) contaminated paddy soil remediation and soil microbial ecology restoration, which presented systematic research about the effect of MFC on soil microbial community and metal resistance. In this study, an SMFC was constructed to remove Cr(VI) in paddy soil and investigate its influence on microbial ecology. The mechanism of Cr(VI) reduction and the change of microbial community structure were comprehensively studied, and the variation of HRGs was also discussed. This research provides a good reference for the microbial remediation of polluted soil and improves the practical application of MFC. In conclusion, the work presented some interesting ideas and new knowledge, which met the journal's scope. However, some shortcomings need to be improved before acceptance for publication. Specific revision suggestions are as follows:**

We would like to thank the reviewers for their time and feedback on this manuscript. Please find our point-to-point responses below.

1. **In the introduction, the author should provide more relevant research on HRGs as appropriate to highlight the innovation of the article.**

**Response:** Thanks for the suggestion.

The change of HRGs in paddy soils is the focus of our discussion and an important innovation point in our research. Widespread antibiotic resistance poses a serious threat to human health. The observed increase in antibiotic-resistant bacteria (ARB) and antibiotic-resistance genes (ARGs) in natural environments has been attributed to the selective pressure generated by the overuse and misuse of veterinary animal feeding and aquaculture [1]. Microorganisms have deployed various strategies to counteract the toxic effects of antibiotics. These include active efflux of the antibiotic

from the microbial cell modification of antibiotic targets and enzymatic modification of the antibiotic [2]. Heavy metals (HMs) accumulate in the environment, and since HMs do not degrade, they create permanent and selective stress on environmental microbes. Even a sub-dose of Cr (especially the Cr(VI) state) can promote plasmid-mediated horizontal gene transfer (HGT) [3], causing enrichment of heavy metal resistance genes (HRGs), and threatening environmental safety [4-7]. Meanwhile, due to the co-selection effect, the long-term existence of HMs also causes the enrichment of antibiotic-resistant bacteria, further increasing the resistance gene spreading risk in the environment [8]. Although the research on ARGs in soil is extensive, the study of MFC operation on HRGs in soil is still very scarce.

In the revised manuscript, we have supplemented the relevant content in the introduction and marked it in red.

Some of the newly added references are as follows:

[1]Ashbolt Nicholas, J., Amézquita, A., Backhaus, T., Borriello, P., Brandt Kristian, K., Collignon, P., Coors, A., Finley, R., Gaze William, H., Heberer, T., Lawrence John, R., Larsson, D.G.J., McEwen Scott, A., Ryan James, J., Schönfeld, J., Silley, P., Snape Jason, R., Van den Eede, C., Topp, E., (2013). Human Health Risk Assessment (HHRA) for Environmental Development and Transfer of Antibiotic Resistance. Environmental Health Perspectives 121(9), 993-1001. https://doi.org/10.1289/ehp.1206316.

[2]van Hoek, A.H., Mevius, D., Guerra, B., Mullany, P., Roberts, A.P., Aarts, H.J., (2011). Acquired Antibiotic Resistance Genes: An Overview. Frontiers in Microbiology 2. https://doi.org/10.3389/fmicb.2011.00203.

[3]Zhang, Y., Gu, A.Z., Cen, T., Li, X., He, M., Li, D., Chen, J., (2018). Sub-inhibitory concentrations of heavy metals facilitate the horizontal transfer of plasmid-mediated antibiotic resistance genes in water environment. Environmental Pollution 237, 74-82. https://doi.org/10.1016/j.envpol.2018.01.032.

[4]Guo, S., Xiao, C., Zhou, N., Chi, R., (2021). Speciation, toxicity, microbial remediation and phytoremediation of soil chromium contamination. Environmental Chemistry Letters 19(2), 1413-1431. https://doi.org/10.1007/s10311-020-01114-6.

[5]Wang, C., Jia, Y., Li, J., Li, P., Wang, Y., Yan, F., Wu, M., Fang, W., Xu, F., Qiu, Z., (2023). Influence of microbial augmentation on contaminated manure composting: metal immobilization, matter transformation, and bacterial response. J. Hazard. Mater. 441, 129762. https://doi.org/https://doi.org/10.1016/j.jhazmat.2022.129762.

[6]Wang, C., Jia, Y., Li, J., Wang, Y., Niu, H., Qiu, H., Li, X., Fang, W., Qiu, Z., (2023). Effect of bioaugmentation on tetracyclines influenced chicken manure composting and antibiotics resistance. Science of The Total Environment 867. https://doi.org/10.1016/j.scitotenv.2023.161457.

[7]Wang, C., Tan, H., Li, H., Xie, Y., Liu, H., Xu, F., Xu, H., (2020). Mechanism study of Chromium influenced soil remediated by an uptake-detoxification system using hyperaccumulator, resistant microbe consortium, and nano iron complex. Environ. Pollut. 257, 113558. https://doi.org/10.1016/j.envpol.2019.113558.

[8]Men, C., Liu, R., Xu, F., Wang, Q., Guo, L., Shen, Z., (2018). Pollution characteristics, risk assessment, and source apportionment of heavy metals in road dust in Beijing, China. Science of The Total

Environment 612, 138-147. https://doi.org/https://doi.org/10.1016/j.scitotenv.2017.08.123.

**2. Some writing details need attention. For example, the subtitle of a paper is sorted incorrectly. Such as section 2.4.**

**Response:** Thank you for your careful examination. During revision, we will carefully review and revise the chapter order to ensure its accuracy.

The original title "2.6 Microbial response during operation " is amended to "2.5 Microbial response during operation "; The original title "2.6.1 Soil biochemical response " is amended to "2.5.1 Soil biochemical response "; The original title "2.6.2 Microbial community structure " is amended to "2.5.2 Microbial community structure "; The original title "2.6.3 HRG Fluctuation " is amended to "2.5.3 HRG Fluctuation "; The original title "2.7 Data analysis " is amended to "2.6 Data analysis ";

**3. Lines 77-78, please explain why the author chose to add 118.8 mg/kg chromium to the soil.**

**Response:** Thank you for your kind suggestion.

In the current study, the Cr concentration in the soil was set referring to the Chinese Soil Environmental Quality Standard (GB15618-2018). The standard stimulated that the first-level risk value of heavy metal risk control in agricultural land soil (if it exceeds the value, there will be soil ecological environment pollution risk) is between 90-150 mg/kg for Cr. After referring to previous studies, 120.0 mg/kg of Cr was spiked in the soil and the final value was determined as 118.8 mg/kg by the FAAS [1-3].

Some of the revised paragraphs are as follows:

[1]Li, Y., Lin, J., Wu, Y., Jiang, S., Huo, C., Liu, T., Yang, Y., Ma, Y., (2024). Transformation of exogenous hexavalent chromium in soil: Factors and modeling. Journal of Hazardous Materials 480, 135799. https://doi.org/https://doi.org/10.1016/j.jhazmat.2024.135799.

[2]Liu, S., Pu, S., Deng, D., Huang, H., Yan, C., Ma, H., Razavi, B.S., (2020). Comparable effects of manure and its biochar on reducing soil Cr bioavailability and narrowing the rhizosphere extent of enzyme activities. Environment International 134, 105277. https://doi.org/https://doi.org/10.1016/j.envint.2019.105277.

[3]Mandal, S., Sarkar, B., Bolan, N., Ok, Y.S., Naidu, R., (2017). Enhancement of chromate reduction in soils by surface modified biochar. Journal of Environmental Management 186, 277-284. https://doi.org/https://doi.org/10.1016/j.jenvman.2016.05.034.

**4. In section 2.5.3, it is hoped that the author can provide specific operations on " The abundance of HRGs in the surface soil of SMFC and OMFC anode after the operation was analyzed using an SYBR Green real-time fluorescence quantitative PCR system ".**

**Response:** Thank you for your professional advice. In the revised manuscript, the related paragraph was revised as:

Microbial DNA Rapid extraction kit (Shenggong Bioengineering Co., LTD., Shanghai, China) was used to extract total DNA from fresh samples. The abundance of HRGs in the surface soil of SMFC

and OMFC anode after operation was analyzed using a SYBR Green real-time fluorescence quantitative PCR system (7500, Thermo Fisher, USA) [1]. The soil of OMFC was used for comparison. The detected genes included HRGs (*chrA, chrB, chrR, recG, nfsA, zupT, fpvA*) and MGEs (*intI, tnpA02, tnpA04, tnpA05*). The primer sequences are provided in Table S2. The specific detection steps were as follows: pre-denaturation at 95°C for 30 s, denaturation at 95°C for 5 s, annealing and extension at 60°C for 30 s. Forty cycles were performed to make three replicates, and 16S rRNA was used as the internal reference gene. The relative gene expression results were analyzed using the $2^{\wedge}(-\Delta\Delta Ct)$ method, which is commonly used for relative quantification, where $\Delta\Delta Ct$ = (Ct target gene - Ct internal reference gene) experimental group - (Ct target gene - Ct internal reference gene) control group.

The specific DNA extraction procedures are provided in Section 4 of the Supplementary Materials.

Some of the newly added references are as follows:

[1]Wang, C., Jia, Y., Li, J., Li, P., Wang, Y., Yan, F., Wu, M., Fang, W., Xu, F., Qiu, Z., (2023). Influence of microbial augmentation on contaminated manure composting: metal immobilization, matter transformation, and bacterial response. J. Hazard. Mater. 441, 129762. https://doi.org/https://doi.org/10.1016/j.jhazmat.2022.129762.

**5. Lines 127-137, for the determination method of Microbial community structure, it is suggested that the author supplement the relevant references.**

**Response:** Thank you for your helpful advice. We have supplemented the methods for determining microbial community structure in Section 2.5.2 with references. Some of the newly added references are as follows:

[1] Bokulich, N. A., Kaehler, B. D. , Ram, R. J. , Matthew, D. , Evan, B. , & Rob, K. , et al. (2018). Optimizing taxonomic classification of marker-gene amplicon sequences with qiime 2's q2-feature-classifier plugin. Microbiome, 6(1), 90.

[2] Deng, Y., Jiang, Y. H., Yang, Y., He, Z., Luo, F., and Zhou, J. (2012). Molecular ecological network analyses. BMC Bioinformatics 13(1), 113.

[3] Faust, K., and Raes, J. (2012). Microbial interactions: from networks to models. Nat Rev Microbiol 10, 538-550.

**6. In lines 181-183, The authors need a more complete explanation for why the voltage of SMFC drops rapidly in the first week and then rises again.**

**Response:** Thank you for your professional advice. In the revised manuscript, the related paragraph is revised as:

Initially, CMFC showed a working circuit voltage (WCV) of 0.55 V and an open circuit voltage (OCV) of 0.68 V (Fig. 2g). In the first week, WCV dropped quickly to 0.45 V but bounced back and stabilized at 0.75 V on day 25, implying the adaptation process of the anode microbes in the soil. We reasonably concluded that in the complex heterogeneous environment of soil, the anodic

EAB needs some time to adapt to fluctuating environmental conditions facing environmental disturbances. During SMFC operation, the anode microbial community could be gradually selected and enriched, and a stable adaptive community is formed, so the SMFC voltage tends to be stable. This conjecture is also reflected in subsequent results.

**7. The Cr shown in both Figure 1e and Figure 1k is easy to be confused. It is recommended to mark them in the appropriate position in the diagram.**

**Response:** Thank you for your kind suggestion.

We have attached clear and eye-catching ICONS to the relevant images in the revised manuscript. The replotted figure is as follows:

[Figure]

**Fig. 1** Characterization of electrode materials before and after operation by SEM and EDS. (a-c) SEM and EDS images of cathode loaded with $Fe_3O_4$; (d-f) SEM and EDS images of cathode after the SMFC operation; (g-i) SEM and EDS images of anode microorganisms; (j-l) SEM and EDS images of the anode after SMFC operation. (Note: The first column is the SEM image, the second

column is the corresponding EDS, and the third column is the corresponding enlarged SEM image.)

8. **Figure 2 shows that the cathode is GF while the anode is not represented by the material name. Please uniformly use the electrode or material noun to provide a good reading experience.**

**Response:** Thanks for the suggestion. We have uniformed the material nouns in the revised manuscript. And Figure 2 is replotted as follows:

[Figure]

**Fig. 2** Characterization of electrode materials. (a-b) Fe2p spectra of cathode/Fe$_3$O$_4$ composite cathode, (c) cyclic voltammetry (CV) curve of cathode/Fe$_3$O$_4$, (d-e) Cr2p spectra of GF composite cathode and Anodic Aluminum foam after operation, (f) XRD spectrum of the cathode-Fe$_3$O$_4$.

[Figure]

**Fig. 3** Power generation performance of SMFC. (a) SMFC output voltage distribution, (b) 15-day vs. 30-day polarization curves and power density curves of the SMFC.

**9. In section 3.4.2, it is suggested that the author briefly explain why the soil biochemical indicators should be determined and what contribution these indicators have to the subject.**

**Response:** Thank you for your kind suggestion.

Soil enzyme activity is an important index to evaluate soil environmental change [1-2]. Soil enzymes, as biocatalysts involved in biochemical reactions, play an important role in nutrient mineralization, decomposition of organic matter, and nutrient cycling. The dynamic measurement of soil enzyme activity in SMFC helps us to understand and analyze the changing state of soil microecology. The interaction analysis of soil enzymes with other indicators will also help us to understand the specific action mechanism of microecology that contributes to Cr reduction. The results showed that the changes in soil enzyme activity were closely related to the changes in soil microbial community structure during SMFC operation, and the changes in the abundance of resistance genes were also reflected.

In the revised manuscript, we have supplemented the relevant content in 3.4.2and marked it in red.

Some of the newly added references are as follows:

[1]Chen, Y., Zuo, M., Yang, D., He, Y., Wang, H., Liu, X., Zhao, M., Xu, L., Ji, J., Liu, Y., Gao, T., (2024). Synergistically Effect of Heavy Metal Resistant Bacteria and Plants on Remediation of Soil Heavy Metal Pollution. Water, Air, & Soil Pollution 235(5), 296. https://doi.org/10.1007/s11270-024-07100-w.

[2]Liu, H., Xu, F., Xie, Y., Wang, C., Zhang, A., Li, L., Xu, H., (2018).ccScience of The Total Environment 645, 702-709. https://doi.org/https://doi.org/10.1016/j.scitotenv.2018.07.115.

**10. In Part 4, give necessary description of Figure 6 "Cr(VI)reduction Mechanism of during SMFC operation".**

**Response:** Thank you for your kind advice. In the revised manuscript, the related paragraph was revised as follows:

Microorganisms have developed efficient detoxification strategies to counteract the toxic effects of heavy metal stress [1-2]. In the SMFC system, Cr(VI) reduction was a synergic result of adsorption/biosorption (including the adsorption of anode and cathode materials, surface catalyst adsorption, and microbial membrane adsorption), bioelectrochemistry reduction, and microbial reduction (including intracellular sequestration, export, reduced permeability, extracellular sequestration, and extracellular detoxification) (Fig. 6).

Some of the newly added references are as follows:

[1] Rouch, D A, Lee, B T O, Morby, A P, 1995. Understanding cellular responses to toxic agents: a model for mechanism-choice in bacterial metal resistance. Journal of Industrial Microbiology 14:132-141.

[2]Tan, H., Wang, C., Zeng, G., Luo, Y., Li, H., Xu, H., (2020). Bioreduction and biosorption of Cr(VI) by a novel Bacillus sp. CRB-B1 strain. Journal of Hazardous Materials 386, 121628. https://doi.org/https://doi.org/10.1016/j.jhazmat.2019.121628

**Reviewer #2:**

**The manuscript entitled " Cr(VI) reduction, electricity production, and microbial resistance variation in paddy soil under microbial fuel cell operation " has been written well and supported with sufficient experimental data. The authors found that during SMFC operation, soil physicochemical properties, enzyme activities, resistance genes, and microbial community structure closely interacted with each other. Although the overall design of experiment and manuscript are fine, some changes are required to be made in manuscript. Therefore, following changes should be included before publication:**

We would like to thank the reviewers for their time and feedback on this manuscript. Please find our point-to-point responses below.

1. **Whether the SMFC has application prospects in practice, whether it can be applied to other soils except rice field soil, and whether the $Fe_3O_4$ will cause secondary pollution to the environment. Its advantage as cathode catalyst should also be mentioned. Please give a brief description.**

**Response:** Thank you for your kind suggestion.

This study presents an innovative treatment strategy that is economical and eco-friendly compared with conventional physicochemical methods. The soil MFC (SMFC) configuration was designed for in situ treatment and allows the anode to utilize the organic contents in sediment without a separator to divide the anode and cathode. Moreover, nutrients are passively supplemented during sedimentation from contaminated water to provide sustainable electron donors for SMFCs. The method can be used for in-situ contaminant treatment in a variety of environments, such as aquaculture ponds, inland lakes, and wetlands [1-2].

The iron oxides ($Fe_3O_4/Fe_2O_3$) are the most commonly used modifier for electrodes. The iron oxide modification mainly improves the kinetics activity of the reaction [3]. Iron-based materials due to the formation of dense flocs with good settling properties that remove phosphorus and other pollutants by adsorption and sweeping, have been widely used in environmental treatment[4]. Iron is a critical component of cytochrome C and iron-sulfur proteins, which are required by most electricigens and play a crucial role in the respiratory chain of microorganisms [5]. Moreover, iron is a large element present in soil and is widely used as a soil conditioner [6]. Therefore, the appropriate amount of $Fe_3O_4$ residue in the soil will not cause secondary pollution.

In the revised manuscript, we have supplemented the relevant content in the introduction and marked it in red.

Some references are as follows :

[1] Zhang, Q., Wang, L., Xu, D., Tao, Z., Li, J., Chen, Y., Cheng, Z., Tang, X., Wang, S., (2023b). Accelerated Pb(II) removal and concurrent bioelectricity production via constructed wetlandmicrobial fuel cell: Structural orthogonal optimization and microbial response mechanism. Journal of Water Process Engineering 56, 104287. https://doi.org/10.1016/j.jwpe.2023.104287.

[2] Youssef, Y.A., Abuarab, M.E., Mahrous, A., Mahmoud, M., (2023). Enhanced degradation of ibuprofen in an integrated constructed wetland-microbial fuel cell: treatment efficiency, electrochemical characterization, and microbial community dynamics. RSC Advances 13(43), 29809-29818. https://doi.org/10.1039/D3RA05729A.

[3] Li, J., Gao, H., (2008). A Renewable Potentiometric Immunosensor Based on Fe3O4 Nanoparticles Immobilized Anti-IgG. Electroanalysis 20(8), 881-887. https://doi.org/https://doi.org/10.1002/elan.200704094.

[4] Venkateswarlu, S., Yoon, M., Kim, M.J., (2022). An environmentally benign synthesis of Fe3O4 nanoparticles to $Fe_3O^4$ nanoclusters: Rapid separation and removal of Hg(II) from an aqueous medium. Chemosphere 286, 131673. https://doi.org/https://doi.org/10.1016/j.chemosphere.2021.131673

[5] Yu, B., Li, Y., Feng, L., (2019). Enhancing the performance of soil microbial fuel cells by using a bentonite-Fe and $Fe_3O_4$ modified anode. Journal of Hazardous Materials 377, 70-77. https://doi.org/https://doi.org/10.1016/j.jhazmat.2019.05.052.

[6] Li, Q., Chang, J., Li, L., Lin, X., Li, Y., (2024). Soil amendments alter cadmium distribution and bacterial community structure in paddy soils. Science of The Total Environment 924, 171399. https://doi.org/https://doi.org/10.1016/j.scitotenv.2024.171399.

**2. Some of the formatting of the manuscript needs to be improved for ease of reading. lines 278-290, the paragraph is too long to grasp the point and requires the author to break it up in appropriate places.**

**Response:** Thank you for your professional advice. In the revised manuscript, the related paragraph was revised as:

At the genus level (Fig. 4c), MFC operation presented a selection effect, with *Terrisporobacter* increasing from 0.81% to 13.71% and *Bacteroides* decreasing from 12.48% to 0.53% in CMFC anode. Compared with the Raw-A, many EABs in CMFC-A decreased, including *Clostridium_sensu_ stricto_5* (from 12.99% to 0.052%), *Clostridium_sensu_stricto_15* (from 4.70% to 0.47%), and *Enterococcus* (from 17.26% to 0.03%).

However, the *Desulfotomaculum* in CMFC-A increased to 3.32% compared with 0.003% in the soil (CMFC-S). Besides, soil indigenous bacteria including *Ramlibacter, Methyloversatilis*, and *Acinetobacter* colonized in the anode and elevated by 4.89~1 579 fold compared with soil. Nevertheless, multiple dominant genera in the soils decreased in CMFC-A than in OMFC-A. For example, *SBR1031, Bacteroidetes_vadinHA17*, and *Anaerolinea* were significantly increased in soil, but less in CMFC-A and OMFC-A. The electric field action to a certain extent helped the anode to resist external microbial intrusion to ensure the stability of the anodic microbial community.

In the revised manuscript, we have supplemented the relevant content in 3.5.2 and marked it in red.

3. **Please explain the sampling process during the SMFC operation and if it will affect the normal operation and cause large errors?**

**Response:** Thank you for your professional advice.

We used a plastic cylindrical straw with a diameter of 0.4 cm and a length of 16 cm as the sediment sampler. The SMFC sediment part is inserted by the sampler vertically at a specific time, and then quickly removed. The upper, middle, and lower parts of the sampler are mixed as a determination sample. And the fresh sample each time is only 8-16 g, only 0.2-0.4% of the SMFC. The total sampling amount shall not exceed 5% of the total SMFC content. As the sampler is much smaller than SMFC, the disturbance is avoided to a great extent and the normal operation of SMFC is guaranteed.

In addition in Section 3 of the Supplementary Materials, we have supplemented the relevant content and marked it in red.

4. **Don't use the notion like 'we' or 'our' etc., as these are the redundant words (not the research words) for the standard journal manuscripts.**

**Response:** Thank you for your kind advice. According to the reviewer's suggestion, the manuscript was carefully revised to make sure the use of research words, and some unnecessary redundant words were removed.

5. **In part 2.5.3 HRG Fluctuation, does the author refer to extracted DNA for HRG detection? Please add the specific steps of extraction.**

**Response:** Thank you for your kind advice. According to the reviewer's suggestion, the details of DNA extraction from the sample were added and marked in red in the supplementary material section 4. The detailed DNA extraction procedure was as follows:

Microbial DNA Rapid extraction kit (Shenggong Bioengineering Co., LTD., Shanghai, China) was used to extract total DNA from fresh samples. Specifically, 0.50 g sample, 0.50 g magnetic beads, and 1.0 ml SLX-Mlus Buffer were added in a 2.0 ml Eppendorf tube, and ground for 250 s under 45 HZ. Then added and mixed with 100 μl DS Buffer, and cultivated under 70 °C for 10 min and then 90°C for 2 min. Then the mixture was centrifuged at 10000 g for 5 min at room temperature. 800 μl supernatant was moved to a new tube and added with 270 μl P2 buffer and 100 μl HTR reagent, and then cultivated under -20 °C for 5 min and then centrifuged again at 10000 g for 5 min. The supernatant was then moved to a new 2 ml tube added with the same amount of XP5 buffer and mixed upside down for 8 min. After magnetic rack adsorption, discard the residual liquid, remove the tube, add 500 μL XP5 Buffer, and mix well. Then adsorbed again with a magnetic rack, discard the residual liquid, remove the tube, add 500 μL PHB, and mix well. Then adsorbed again with a magnetic rack, discard the residual liquid, remove the tube, add 500 μL SPW Wash Buffer, and mix well (repeat this step twice). Then the mixture was adsorbed again with a magnetic rack, discard the residual liquid, centrifuge the tube under 10000 g for 10 s. Then the beads were adsorbed again with a magnetic rack, discard the residual liquid, and let stand for 8 min. After that, the beads were added

with 100 µL elution buffer, mixed, and let stand for 5 min. Finally, after adsorbing with a magnetic rack, the supernatant was moved to a new 1.0 ml Eppendorf tube, and total DNA was obtained for further use.

**6.  For gene extraction was it dried or fresh samples?**

**Response:** Thank you for your careful work. For DNA extraction, 0.50 g fresh sample was used after homogenization. The detailed information was added and marked in red in the material and method section of the revised manuscript.

The related paragraph was revised as:

Microbial DNA Rapid extraction kit (Shenggong Bioengineering Co., LTD., Shanghai, China) was used to extract total DNA from fresh samples (The extraction method is shown in the supplementary material).

**7. Lines 157-159, " After operation, the typical peaks of Cr(III), Cr(VI), and element Cr (576.1 and 578.92) were observed on both electrodes by XPS (Fig. 2d-e) " are insufficient to explain that the reduction and immobilization of Cr(VI) by the electrodes.**

**Response:** Thank you for your careful work.

Chromium exists mainly in the III and VI oxidation states in soil. The peaks at 576.9eV and 579eV are typical peaks of Cr(III) and Cr(VI), respectively. After operation, the typical peaks of Cr(III) and Cr(VI) can be observed on the X-ray energy spectra of both cathode and anode, and Cr(VI) and Cr(III) are present on the cathode and cathode of SMFC. The results showed that at least the valence state of Cr changed on the electrode and Cr(VI) was reduced and fixed by the electrode[1].

In the revised manuscript, we have supplemented the relevant content in 3.1 and marked it in red.

Some references are as follows:

[1]  Kim, C., Lee, C.R., Song, Y.E., Heo, J., Choi, S.M., Lim, D.-H., Cho, J., Park, C., Jang, M., Kim, J.R., (2017). Hexavalent chromium as a cathodic electron acceptor in a bipolar membrane microbial fuel cell with the simultaneous treatment of electroplating wastewater. Chemical Engineering Journal 328, 703-707. https://doi.org/https://doi.org/10.1016/j.cej.2017.07.077.

**Reviewer #3:**

**Niu et al. constructed a soil MFC to reduce Cr(VI) in paddy soil and investigate its impact on the microbial community and microbial resistance traits. The findings provide some important clues supporting the application of bioelectrochemical systems in contaminated paddy soils, offering a potential approach for environmental remediation and detoxification. However, several details require further improvement before being considered for publication.**

We thank the reviewers for their time and feedback on this manuscript. Our point-to-point responses are below.

**Specific comments**

1. **Please explain why aluminum foam was chosen over other materials and what advantages it offers in this experiment.**

**Response:** Thank you for your suggestion.

In this experiment, the anode is operated in a low oxygen moist soil environment, and the soil medium composition is complex, requiring the anode to have high corrosion resistance. Porous metal skeletons, such as foam copper, foam nickel and foam aluminum, are known for their good conductivity and ability to even distribute the metal-ion flux through spatial segmentation[1]. Among them, foam aluminum has high potential as an electrochemical substrate material due to its lightweight, low cost and high abundance [2][3]. The material has a very low density, good electrical conductivity, and resistance to acid and alkali corrosion [4]. In addition, aluminum foam's unique multi-space structure not only helps load more functional bacteria in the early stage but also increases the anode conduct on electron efficiency of SMFC with its excellent specific surface area. Due to the good corrosion resistance, electricity conductivity, and high specific surface area, aluminum foam was chosen as the anode material.

Some references are as follows:

[1] Ding, Y., Zhang, Q., Rui, K., Xu, F., Lin, H., Yan, Y., Li, H., Zhu, J., Huang, W., (2020). Ultrafast Microwave Activating Polarized Electron for Scalable Porous Al toward High-Energy-Density Batteries. Nano Letters 20(12), 8818-8824. https://doi.org/10.1021/acs.nanolett.0c03762.

[2] Chen, J., Wang, Y., Li, S., Chen, H., Qiao, X., Zhao, J., Ma, Y., Alshareef, H.N., (2023). Porous Metal Current Collectors for Alkali Metal Batteries. Advanced Science 10(1), 2205695. https://doi.org/https://doi.org/10.1002/advs.202205695.

[3] Lamiel, C., Hussain, I., Ma, X., Zhang, K., (2022). Properties, functions, and challenges: current collectors. Materials Today Chemistry 26, 101152. https://doi.org/https://doi.org/10.1016/j.mtchem.2022.101152.

[4] Rossi, S., Bizzotto, M., Deflorian, F., Fedel, M., (2019). Study of anodizing process on aluminium foam to improve the corrosion behavior. Surface and Interface Analysis 51(12), 1194-1206. https://doi.org/https://doi.org/10.1002/sia.6610.

**2. In the manuscript, the authors included a virtual model diagram of the SMFC. Please include a realistic photograph of the setup in the subsequent revision to enhance clarity and provide a more comprehensive understanding.**

**Response:** Thanks for the suggestion. We have added a realistic photograph of the SMFC device to the supplementary material. The picture is as follows:

The left part of the picture is the SMFC photo, and the right part is the model groups. A plastic box (140.0×85.0×165.0 mm) was used as the SMFC reactor, with 1.50 kg soil and overlying water of 3.0 cm to simulate the flooded state during rice planting. The cathode was floated on the water surface while the anode was buried (about 3.0 cm from the bottom). The cathode and anode were connected to a 2000 Ω resistor using titanium wire. The water level was kept constant by daily replenishment.

[Figure]

**Fig. S1** SMFC structure and experimental grouping.

**3. In the abstract, the authors state: 'Fe₃O₄ nanoparticle as the cathodic catalyst effectively boosted power generation (0.97 V, 102.0 mW/m²), whose porous structure and reducibility also contributed to Cr reduction and immobilization.' However, there is a lack of introduction regarding the cathode catalysts.**

**Response:** Thank you for your professional advice. In the revised manuscript, the related paragraph was revised as:

The iron oxides ($Fe_3O_4$/$Fe_2O_3$) are most commonly used as the modifier. The iron oxide modification mainly improves the kinetics activity of the reaction[1]. Iron-based materials due to the formation of dense flocs with good settling properties that remove phosphorus and other pollutants by adsorption and sweeping, have been widely used in environmental treatment[2]. Iron is a critical component of cytochrome C and iron-sulfur proteins, both of which are required by most electricigens and play a crucial role in the respiratory chain of microorganisms[3].

Some of the newly added references are as follows:

[1] Li, J., Gao, H., (2008). A Renewable Potentiometric Immunosensor Based on Fe3O4 Nanoparticles Immobilized Anti-IgG. Electroanalysis 20(8), 881-887. https://doi.org/https://doi.org/10.1002/elan.200704094.

[2] Yu, B., Li, Y., Feng, L., (2019). Enhancing the performance of soil microbial fuel cells by using a bentonite-Fe and Fe$_3$O$_4$ modified anode. Journal of Hazardous Materials 377, 70-77. https://doi.org/https://doi.org/10.1016/j.jhazmat.2019.05.052.

[3] Venkateswarlu, S., Yoon, M., Kim, M.J., (2022). An environmentally benign synthesis of Fe3O4 nanoparticles to Fe$_3$O$_4$ nanoclusters: Rapid separation and removal of Hg(II) from an aqueous medium. Chemosphere 286, 131673. https://doi.org/https://doi.org/10.1016/j.chemosphere.2021.131673

**4. The manuscript mentions the source and size of anode and cathode materials in section 2.2.2 Electrodes Preparation. However, some essential characteristics, such as the porosity and bulk density of the materials, are missing. Including these details is necessary to provide a comprehensive understanding of the electrode properties.**

**Response:** Thank you for your professional advice. In the revised manuscript, the related paragraph was revised as:

Aluminum foam (66.0×54.0×5.0 mm, porosity 60-80%, bulk density 0.50-1.10 g/cm$^3$) (SANZHENG Metal material, Chengdu, China) was used as anode. The anode microflora was derived from municipal sludge (Chengdu Sixth Sewage Treatment Plant, China) after acclimating with 100 mg/L Cr(VI). Before assembling, the aluminum foam was cultivated in the anode microflora for 2 weeks. Then the anode was tied to titanium mesh tightly with titanium wire. Graphite felt (GF) (100.0×50.0×3.0 mm, bulk density 0.10-0.15 g/cm$^3$) was used as the cathode (Table S1). Before use, it was cleaned, dried, and loaded with Fe$_3$O$_4$ as the ORR catalyst, as detailed in section 1 of the supplementary material. For characterization, we utilized a scanning electron microscope (SEM) to examine the structure and morphology of the electrode surface. In addition, we performed X-ray photoelectron spectroscopy (XPS) and energy dispersive spectroscopy (EDS) to analyze the valence state and element composition. The phase composition was determined using an X-ray diffractometer (XRD).

**5. Lines 138-143: Please provide detailed information on HRGs determination, including the reagents, instruments, and procedures used.**

**Response:** Thank you for your professional advice. In the revised supplementary material, the related paragraph was added in the section 4:

Microbial DNA Rapid extraction kit (Shenggong Bioengineering Co., LTD., Shanghai, China) was used to extract total DNA from fresh samples. Specifically, 0.50 g sample, 0.50 g magnetic beads, and 1.0 ml SLX-Mlus Buffer were added in a 2.0 ml Eppendorf tube, and ground for 250 s under 45 HZ. Then added and mixed with 100 μl DS Buffer, and cultivated under 70 ℃ for 10 min and then 90℃ for 2 min. Then the mixture was centrifuged at 10000 g for 5 min under room temperature. 800 μl supernatant was moved to a new tube and added with 270 μl P2 buffer and 100 μl HTR reagent, and then cultivated under -20 ℃ for 5 min and then centrifuged again at 10000 g for 5 min. The supernatant was then moved to a new 2 ml tube and added with the same amount of XP5 buffer and mixed upside down for 8 min. After magnetic rack adsorption, discard the residual liquid, remove the tube, add 500 μL XP5 Buffer, and mix well. Then adsorbed again with a magnetic rack, discard the residual liquid, remove the tube, add 500 μL PHB, and mix well. Then adsorbed again with a magnetic rack, discard the residual liquid, remove the tube, add 500 μL SPW Wash Buffer, and mix well (repeat this step twice). Then the mixture was adsorbed again with a magnetic

rack, discard the residual liquid, centrifuge the tube under 10000 g for 10 s. Then the beads were adsorbed again with a magnetic rack, discard the residual liquid, and let stand for 8 min. After that, the beads were added with 100 μL elution buffer, mixed, and let stand for 5 min. Finally, after adsorbing with a magnetic rack, the supernatant was moved to a new 1.0 ml Eppendorf tube, and total DNA was obtained for further use. The PCR reaction system was constructed.

**6. Please check typos throughout the text. As an example, supplementary material Table S5, 'conFig.urations'?**

**Response:** Thank you for your careful examination. During revision, we will carefully review and revise the typos to ensure its accuracy.

The "conFig.urations " will be amended to "configurations ".

**7. Some figures in the manuscript need to be improved.**

**Line 167: The annotations in Figure 1 are too small to be clearly seen.**

**Line 172: the icon in Figure 2 is blurred, and the the resolution need to be improved.**

**Response:** Thank you for your kind suggestion. We have attached clear and eye-catching ICONS to the relevant images in the revised manuscript.

[Figure]

**Fig. 1** Characterization of electrode materials before and after operation by SEM and EDS. (a-c) SEM and EDS images of cathode loaded with $Fe_3O_4$; (d-f) SEM and EDS images of cathode after the SMFC operation; (g-i) SEM and EDS images of anode microorganisms; (j-l) SEM and EDS images of the anode after SMFC operation. (Note: The first column is the SEM image, the second column is the corresponding EDS, and the third column is the corresponding enlarged SEM image.)

[Figure]

**Fig. 2** Characterization of electrode materials. (a-b) Fe2p spectra of cathode/Fe$_3$O$_4$ composite cathode, (c) cyclic voltammetry (CV) curve of cathode/Fe$_3$O$_4$, (d-e) Cr2p spectra of GF composite cathode and Anodic Aluminum foam after operation, (f) XRD spectrum of the cathode-Fe$_3$O$_4$.

[Figure]

**Fig. 3** Power generation performance of SMFC. (a) SMFC output voltage distribution, (b) 15-day vs. 30-day polarization curves and power density curves of the SMFC.